theoretical biology/computer modelling and simulation/statistics

open science, funding, replication, reproducibility, cultural evolution

**Author for correspondence:**
Paul E. Smaldino
e-mail: paul.smaldino@gmail.com

# Open science and modified funding lotteries can impede the natural selection of bad science

Paul E. Smaldino[1], Matthew A. Turner[1]
and Pablo A. Contreras Kallens[2]

[1]Department of Cognitive and Information Sciences, University of California, Merced, CA, USA
[2]Department of Psychology, Cornell University, Ithaca, NY, USA

PES, 0000-0002-7133-5620

Assessing scientists using exploitable metrics can lead to the degradation of research methods even without any strategic behaviour on the part of individuals, via 'the natural selection of bad science.' Institutional incentives to maximize metrics like publication quantity and impact drive this dynamic. Removing these incentives is necessary, but institutional change is slow. However, recent developments suggest possible solutions with more rapid onsets. These include what we call *open science improvements*, which can reduce publication bias and improve the efficacy of peer review. In addition, there have been increasing calls for funders to move away from prestige- or innovation-based approaches in favour of lotteries. We investigated whether such changes are likely to improve the reproducibility of science even in the presence of persistent incentives for publication quantity through computational modelling. We found that modified lotteries, which allocate funding randomly among proposals that pass a threshold for methodological rigour, effectively reduce the rate of false discoveries, particularly when paired with open science improvements that increase the publication of negative results and improve the quality of peer review. In the absence of funding that targets rigour, open science improvements can still reduce false discoveries in the published literature but are less likely to improve the overall culture of research practices that underlie those publications.

# 1. Introduction

The 'natural selection of bad science' refers to the degradation of research methodology that results from the hiring and

promotion of scientists on the basis of quantitative metrics. It occurs when those metrics—such as publication count and journal impact factor—become decoupled from the qualities of research they are intended to measure [1]. The persistence of poor research methods is a serious concern. They can lead to widespread false discoveries, wasted effort, and possibly even lost lives in fields such as medicine or engineering where poorly informed decisions can have dire consequences.

The idea that incentives for the quantity and impact factor of publications harms science is not new. Many concerns have focused on the strategic, purposeful and self-interested adoption of questionable research practices by scientists [2–6]. Let us assume that more successful individuals preferentially transmit their methods and perspectives (cf. [7]). If career success is linked to high-volume output and higher output is in turn correlated with reduced rigour, then methodology will worsen even if no one actively changes their behaviour strategically. This dynamic requires only that there are bottlenecks in the hiring and promotion of scientists and that success in traversing those bottlenecks is associated with quantitative metrics that may be exploited.

Competition for permanent jobs in academic science is fierce. A recent study found that the ratio of newly awarded PhDs to open tenure track positions in a given year is approximately five to one in anthropology [8], with similar ratios found in the biomedical sciences [9]. In general, the number of open faculty positions in STEM fields amounts to only a small fraction of the number of total PhDs awarded each year [10,11]. Although not all PhDs seek out academic positions, such positions remain highly desirable and there are reliably many more individuals vying for any given position than there are available spots. Selection at these bottlenecks is non-random. Success is associated with particular features, called selection pressures in evolutionary theory, that influence whether or not an individual traverses the bottleneck. In academic science, this pressure is often linked to the publication history of the particular individual, as evinced by the clichéd admonition to 'publish or perish.'

The pressure to publish has a long history in academia—the use of the English phrase 'publish or perish' dates at least as far back as the 1940s [12]. However, evidence suggests that this pressure may be increasing. Brischoux & Angelier [13] found that the number of publications held by evolutionary biologists at the time of hiring at the French institution CNRS doubled between 2005 and 2013. Zou et al. [14] studied psychologists across subfields in the USA and Canada, and found that new assistant professors hired between 2010 and 2015 averaged 14 publications at the time of hiring, compared with an average of less than seven publications for first-year postdocs. This indicates that substantially more output is required than is typical at the time of graduation from a doctoral program. Focusing only on cognitive psychologists in Canada, Pennycook & Thompson [15] showed that while newly hired assistant professors averaged 10 publications in 2006–2011, this had increased to 28 publications by 2016. A large machine learning study of over 25 000 biomedical scientists showed that, in general, successful scientists end up publishing substantially more papers and in higher-impact journals than those researchers who end up leaving academia [16].

For many scientists and policy makers, it is not obvious that selection for productivity and journal impact factor are bad things. Indeed, it seems that we should *want* our scientists to be productive and for their work to have a wide impact. The problem is that productivity and impact are in reality quite multifaceted but are often assessed with crude, quantitative metrics like paper count, journal impact factor and *h*-index. As Campbell [17] (p. 49) long ago noted, 'The more any quantitative social indicator is used for social decision-making, the more subject it will be to corruption pressures and the more apt it will be to distort and corrupt the social processes it is intended to monitor.' And as shown by Smaldino & McElreath [1], such an incentive-driven mechanism can be damaging even when all actors are well-meaning.

The computational model presented by Smaldino & McElreath [1] made several pessimistic—if realistic—assumptions about the ecosystem of academic science. We focus on two. First, it was assumed that publishing negative or null results is either difficult or, equivalently, confers little or no prestige.[1] Second, it was assumed that publishing novel, positive results is always possible. In other words, the model ignores the corrective role of peer review or, equivalently, assumes it is ineffective. In addition to these two assumptions, the study also assumed that the rate at which research groups could produce results was limited only by the rigour of their methods. However, empirical research very often requires external funding. Grant agencies, therefore, have the power to shape the type of science that is produced by adjusting the criteria on which they allocate funding. In particular, recent

---

[1]The equivalence stems from the fact that it may matter little whether negative results are published—at least in terms of how selection acts on methodology—if they are not weighted similarly to positive results in decisions related to hiring and promotion.

calls for lottery-based funding allocation [18–22] deserve a closer look with regard to their potential influence on reproducible science.

Changing the selection pressures for publication quantity and impact will be an arduous task. Large-scale institutional change is slow. Indeed, it is likely a design feature of institutions that they are hard to change. In his seminal treatise on institutions, Douglass North [23] (p. 83) noted: 'Change typically consists of marginal adjustments to the complex of rules, norms, and enforcement that constitute the institutional framework. The overall stability of an institutional framework makes complex exchange possible.' In academic science, the essential task of changing the norms and institutions regarding hiring, promotion, and publication is likely to be difficult and slow-going.

Here, we explore how a more limited set of changes to the cultural norms of academic science might ameliorate the pernicious effects of the aforementioned hiring bottleneck. Specifically, we investigate the influence of three key factors on the natural selection of bad science: publication of negative results, improved peer review, and criteria for funding allocation. Changes to the publication of negative results and the quality of peer review can be driven by flagship journals and scientific societies, which can adopt or introduce new policies with relative speed. For example, there is an increasing number of journals with mandates to accept all well-done studies regardless of the perceived importance of the results, thereby reducing publication bias. These include *PLoS ONE, Collabra, PeerJ*, and *Royal Society Open Science*. In just the last few years, much progress has been made on these fronts. Such progress is often associated with what is sometimes called the Open Science movement [24], and so for convenience we refer to reduced publication bias and improved peer review as *open science improvements*, though it is, of course, possible to support these improvements without any ideological buy-in. Changes to funding criteria can also be rapid, as funding agencies can act unilaterally to influence what scientific proposals are enabled. For convenience, we refer to the union of open science and funding improvements as *rapid institutional changes* to denote that they are can be implemented quickly, at least relative to the time scale needed to remove the emphasis on quantitative metrics at the hiring and promotion stages.

By investigating the long-term consequences of these more rapidly implemented changes, we aim to infer the extent to which the recent and proposed changes to the culture of science help to make subsequent science better and more reproducible. We do so by studying an evolutionary model of scientific ecosystems that further develops previous models of science [1,25]. Before describing the model, we review the proposed changes we examine, the rationale behind these changes, and the key questions regarding their consequences.

# 2. Rapid institutional changes

## 2.1. Publishing of negative results

In the model of Smaldino & McElreath [1], it was assumed that negative results were rarely or never published, even though consistent publication of negative results has been shown to dramatically increase the information quality of the published literature [25,26]. This was a reasonable assumption because negative results—results that fail to advance a new or novel hypothesis—are rarely published [27], and indeed are rarely even written up for submission [28].

Recently, however, publication of negative results has been encouraged and applauded in many circles. As mentioned, journals are increasingly willing to publish such papers. More and more journals also accept registered reports, in which a research plan is peer reviewed before a study is conducted. Once approved, the paper's acceptance is contingent only on adherence to the submitted plan, and not on the character of the results [29,30]. A recent study by Allen & Mehler [31] found that papers resulting from registered reports exhibited much higher rates of negative results than in the general literature. Of course, even if negative results are published, they may not have the same status as novel results. If negative results are publishable and worthy of prestige, the question is: *How common or prestigious must the publication of negative results be, relative to positive results, in order to mitigate the natural selection of bad science?*

## 2.2. Improving peer review

In the model of Smaldino & McElreath [1], it was assumed that publishing novel, positive results was always possible, ignoring the corrective role of peer review. This was a simplifying assumption, but

probably a reasonable one. There have recently been many demonstrations of failed replications of peer-reviewed papers that were originally published in reputable journals, including in the biomedical [32,33], psychological [34] and social sciences [35]. This indicates limitations to the ability of reviewers to weed out incorrect results.[2] Moreover, reviewers are hardly objective. When reviews are not double-blind, reviewers may be more likely to accept papers by high status individuals and less likely to accept papers by women and minority scientists[3] [36,37]. This may reflect a more general trend whereby reviewers are more likely to accept results that fit with their pre-existing theoretical perspectives [38]. The inefficacy of peer review is further illustrated by recent studies showing low inter-reviewer reliability—papers submitted to the same journal or conference—may be accepted by one set of reviewers and rejected by another. Indeed, many studies have found low correlation between reviewer decisions on grant panels [39–41], conference proceedings [42,43] and journals [44–46]. While we certainly do not expect peer reviewers to ever be able to perfectly weed out incorrect results, the evidence indicates that peer reviewers are not effectively optimizing their reviews for truth or methodological rigour.

Recently, however, there have been advances leading to improved peer review. Registered reports remove biases based on the novelty of a study's results [29,30]. Double-blind peer review aims to reduce biases further [37,47,48]. Journals increasingly require or incentivize open data and methods (including making available the raw data, analysis scripts and model code), which improves the ability of peer reviewers to assess results, and the increased use of repositories such as OSF and GitHub facilitates this behaviour (though these journals prescriptions are still not perfectly effective; see [49,50]). Open peer review and the increased use of preprint servers also allow for a greater number of critical eyes to read and comment on a manuscript before it is published [51,52]. Finally, better training in statistics, logic, and best research practices—as evidenced by the popularity of books, MOOCs, podcasts, symposia, and conferences on open science—likely reflects increased awareness of the problems in science, which may make reviewers better. For example, the software package statcheck scans papers for statistical tests and flags mathematical inconsistencies, and has been used in psychological research to improve statistical reporting [53,54]. Of course, peer review serves several important functions beyond its corrective role in reducing false discovery, including improving the precision of writing, suggesting clarifying analyses, and connecting work with relevant literature.[4] For simplicity, we focus only on its corrective role.

If reviewers were to be better able to prevent poorly performed studies or erroneous findings from being published, the question is: *How effective does peer review have to be to mitigate the natural selection of bad science?*

## 2.3. Funding allocation

The model of Smaldino & McElreath [1] made no explicit assumptions about the influence of funding on research productivity. Rather, it was assumed that the rate at which research groups could produce results was limited only by the rigour of their methods. However, research in most scientific fields requires funding, and so funders have tremendous leverage to shape the kind of science that gets done by providing the resources that allow or stymie research [56]. The weight of fund allocation is such that a researcher who is unsuccessful at securing funding may end up losing their academic position [57]. Funding decisions can be made quickly, and can therefore rapidly change the landscape of research. For example, several funding agents—including the Gates Foundation, the state of California, and the entire Science Europe consortium—require all funded research to be published in Open Access journals [58–60]. Changes to the criteria used to assess research proposals may have dramatic long-term effects on the scientific research that is performed and published. Through them, funding agencies could reinforce or counter the effects of hiring bottlenecks. If agencies prioritize funding individuals with records of high productivity, for example, the pressure for reducing rigour in exchange for increased yield will continue beyond hiring and be exacerbated throughout a scientist's career. On the other hand, if agencies prioritize methodological rigour, they might be able to reduce the detrimental effects of the same hiring bottleneck.

---

[2]Editors at high prestige journals are also rumoured to accept papers that are unlikely to be true but *are* likely to be newsworthy [4].

[3]In practice, even double-blind review cannot ensure that reviewers will not discover a paper's authors, but it probably helps.

[4]There is also a dark side to peer review, to which anyone who has faced the dreaded 'Reviewer 2' can attest. At worst, it can serve to impede spread of innovative practices or theories that contradict prevailing paradigms [55] (Gil-White, F. Academic market structure and the demarcation problem: science, pseudoscience and a possible slide between. Unpublished manuscript.).

Little is known about how funding criteria influence the replicability of the funded research. Our question is: *How does the criterion on which research funds are allocated influence the natural selection of bad science?*

In reality, the criteria used by funders to allocate grants are multidimensional and complex, and attempt to take into account aspects like novelty, feasibility and reputation [61]. For simplicity, these nuances are not considered here. Instead, we focus our analysis on three extreme strategies for funding allocation: publication history (PH), methodological integrity (MI) and random allocation (RA), described in greater detail below. These strategies award funds based on which lab has the most publications, the lowest false positive rate (i.e. the most rigorous methods), or completely at random. For convenience, we refer to these three strategies as 'pure' strategies, because they either maximize a single function or are completely random. We will also consider hybrid strategies that combine aspects of RA and MI, including modified lotteries.

### 2.3.1. Publication history

Funders allocate based on the previous publication history of the research groups in question. This models a reputational effect that reinforces the selection criteria assumed to be at work in the hiring process, such that those who are able to publish at higher volumes are also best able to secure funding.

### 2.3.2. Methodological integrity

Ideally, funding agencies want to fund research that is reliable and executed with rigour and integrity. Of course, integrity is difficult to assess. If we could accurately and easily assess the precise quality of all labs and dole out rewards and punishments accordingly, improving science would be rather straightforward [62]. Nevertheless, it is often the case that at least *some* information about the integrity of a research lab is available, perhaps via reputation and peer assessment of prior work. Our focus on MI here may be viewed as an ideal 'best case' scenario, as well as a measuring stick by which to regard the other funding allocation strategies considered.

### 2.3.3. Random allocation

Recently, a number of scholars and organizations have supported a type of lottery system for allocating research funds [18–22]. There are many appealing qualities about such a funding model, including (i) it would promote a more efficient allocation of researchers' time [22]; (ii) it would likely increase the funding of innovative, high-risk-high-reward research [18,21]; and (iii) it would likely reduce gender and racial bias in funding, and reduce systematic biases arising from repeat reviewers [21]. Such biases can lead to cascading successes that increase the funding disparity between those who, through luck, have early successes, and those that do not [63]. There may also be drawbacks to such a funding model, including increased uncertainty for large research groups that may be disproportionately harmed by any gaps in funding. Regardless, most previous research on alternative funding models has ignored their influence on the quality and replicability of published science, which is our focus.

### 2.3.4. Hybrid strategies

Most serious calls for funding lotteries have proposed that a baseline threshold for quality must first be met in order to qualify projects for consideration in the lottery. The pure RA strategy ignores this threshold. We, therefore, also consider two hybrid funding strategies that combine aspects of RA and MI. The first of these is a mixed strategy (MS) that allocates funds using the MI strategy a proportion $X$ of the time and the RA strategy for the remainder. The second is a *modified lottery* (ML), which allocates funds randomly but excludes any labs with a false positive rate above a threshold $A$. We will show that such hybrid strategies, which are more realistically implemented than either of their constituent pure strategies, are quite effective at keeping false discovery rates low.

## 3. Model

Our model extends the model presented by Smaldino & McElreath [1]. We consider a heterogeneous population of $n$ labs, each of which varies in its methodological rigour. The labs will investigate new hypotheses if they have sufficient funds and then attempt to publish their results. Older labs are

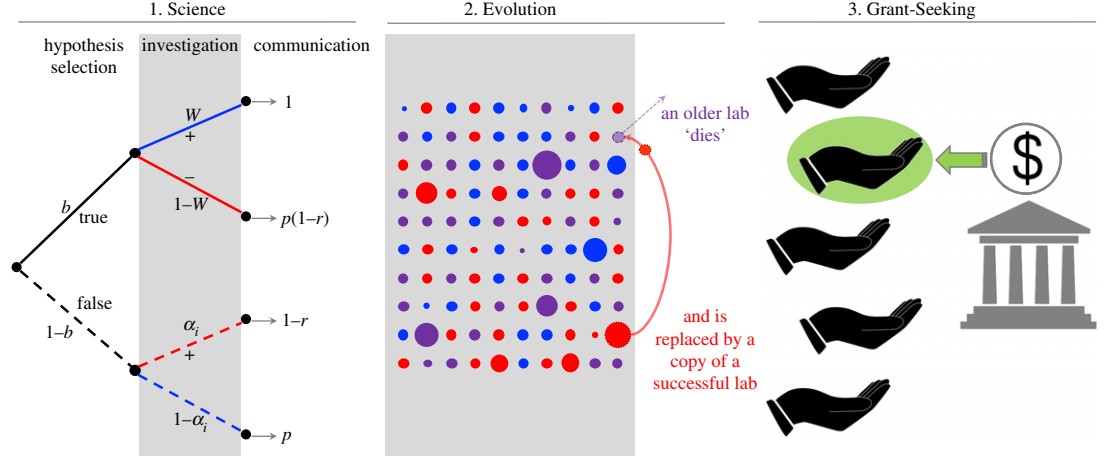

**Figure 1.** Schematic of the model dynamics in three stages: Science, Evolution and Grant-Seeking. In the Science stage (1), a hypothesis is either true (solid lines) or false (dashed lines). When investigated, the results are either positive or negative (results congruent with the true epistemic state of the hypothesis are indicated by blue, results incongruent are indicated by red). Results are then communicated with a probability influenced by the publication rate of negative results ($p$) and the efficacy of peer review ($r$). In the Evolution stage (2), labs vary by their methodological rigour (indicated by arbitrary colour) and publication history (indicated by size). At each time step, one of the older labs 'dies,' and is replaced by a new lab that inherits its methods from among the most productive extant labs. In the Grant-Seeking stage (3), a subset of labs apply for funding, which is awarded to the lab that best meets the criterion used by the funding agency.

gradually removed from the simulation as they 'retire,' and new labs arise by inheriting the methods of successful older labs—that is, labs who have published many papers. The dynamic is one of cultural evolution (cf. [1,64,65]), and represents the idea that, in many disciplines, highly productive labs are more likely to be the source of new PIs.

The rigour of each lab $i$ is represented by a single term, $\alpha_i$, which is the intrinsic false positive rate of studies conducted in that lab. At the beginning of each run, all labs are initialized with $\alpha_0 = 0.05$. The rate at which labs can perform new studies is constrained by whether or not the lab has funding. Each lab is initialized with $G_0 = 10$ funds (which may be thought of as 'startup funds'), such that it costs 1 unit of funding to conduct a new study. Additional funds can only be acquired by applying for a grant.

The dynamics of the model proceed in discrete time steps, each of which consists of three stages: *Science*, *Evolution* and *Grant-Seeking* (figure 1). In the Science stage, each lab with sufficient funds has the opportunity to select a hypothesis, investigate it experimentally, and attempt to communicate their results through peer-reviewed publication. Hypotheses are assumed to be strictly true or false, though their correct epistemic states cannot be known with certainty but can only be estimated using experiments. This assumption is discussed at length in McElreath & Smaldino [25]. In the Evolution stage, an existing lab 'dies' (ceases to produce new research), making room in the population for a new lab that adopts the methods of a progenitor lab. More successful labs are more likely to produce progeny. In the Grant-Seeking stage, labs have the opportunity to apply for funds, which are allocated according to the strategy used by the funding agency. We describe these stages in more detail below. Values and definitions for all parameter values are given in table 1.

## 3.1. Science

The Science stage consists of three phases: hypothesis selection, investigation and communication. Every time step, each lab $i$, in random order, begins a new investigation if and only if it has research funds greater than zero. If a new experimental investigation is undertaken, the lab's research funds are reduced by 1, and the lab selects a hypothesis to investigate. The hypothesis is true with probability $b$, the base rate for the field.[5] It is currently impossible to accurately calculate the base rate in most

---

[5]In reality, the base rate reflects the ability of researchers to select true hypotheses, and thus should properly vary between labs. Because our analysis focuses on methodological rigour and not hypothesis selection, we ignore this inter-lab variation. In our opinion, better hypothesis selection stems at least in part from stronger engagement with rich and formalized theories, such as we attempt to provide here.

**Table 1.** Summary of parameter values used in computational experiments.

| parameter | definition | values tested |
| --- | --- | --- |
| $n$ | number of labs | 100 |
| $b$ | base rate of true hypotheses | 0.1 |
| $W$ | power of experimental methods | 0.8 |
| $\alpha_0$ | initial false positive rate for all labs | 0.05 |
| $G_0$ | initial funds for all labs | 10 |
| $G$ | funding per grant | $\{10, 35, 60, 85\}$ |
| $d$ | number of labs sampled for death, birth and funding events | 10 |
| $\epsilon$ | standard deviation of $\alpha$ mutation | 0.01 |
| $r$ | efficacy of peer review | $\{0, 0.1, \ldots, 1\}$ |
| $p$ | publication rate for negative results | $\{0, 0.1, \ldots, 1\}$ |
| $S$ | funding allocation strategy | $\{PH, RA, MI\}$ |
| $X$ | proportion of funds allocated to most rigorous labs under mixed funding | $\{0, 0.1, \ldots, 1\}$ |
| $A$ | minimum false positive rate for funding under modified lottery | $\{0.1, 0.2, \ldots, 1\}$ |

experimental sciences; it may be as high as 0.1 for some fields, but it is likely to be much lower in others [25,66–68]. For all results presented here, we use a fairly optimistic $b = 0.1$. Some researchers have claimed to us in personal communications that they believe their own base rates to be substantially higher. Whatever the veracity of such claims, we have repeated our analyses with $b = 0.5$ in the electronic supplementary material, appendix and obtain qualitatively similar results, albeit with predictably lower false discovery rates in all conditions.

All investigations yield either positive or negative results. A true hypothesis yields a positive result with probability $W$, representing the power of the methods used by each group, $\Pr(+\,|T)$. For simplicity, and to explore a fairly optimistic scenario, we fix the power to a relatively high value of $W = 0.8$. A false hypothesis yields a positive result with probability $\alpha_i$, which reflects the lab characteristic methodological rigour. It is worth noting that in the model of [1], increased rigour not only yielded fewer false positives but also decreased the rate at which labs could produce new results and thereby submit new papers. Here, we disregard this assumption in the interest of tractability. Adding a reduction in productivity in response to rigour is likely to decrease the improvements from rapid institutional change. However, a theoretically motivated reason to ignore reduced productivity is an inherent difficulty in calibrating the *extent* to which such a reduction would manifest.

Upon obtaining the results of an investigation, the lab attempts to communicate them to a journal for publication. This is where open science improvements come into play. We assume that positive results are always publishable, while negative results are publishable with rate $p$. Larger $p$ represents a reduction in publication bias. Moreover, effective peer review can block the publication of erroneous results—i.e. a positive result for a false hypothesis or a negative result for a true hypothesis. Such results are blocked from publication with probability $r$, representing the efficacy of peer review.[6] Figure 1 illustrates these dynamics. We keep track of the total number of publications produced by each lab.

## 3.2. Evolution

Once all labs have had the opportunity to perform and communicate research, there follows a stage of selection and replication. First, a lab is chosen to die. A random sample of $d$ labs is obtained, and the oldest lab of these is selected to die, so that age correlates coarsely but not perfectly with fragility. If multiple labs in the sample are equally old, one of these is selected at random. The dying lab is then removed from the population. Next, a lab is chosen to reproduce. A new random sample of $d$ labs is obtained, and from among these the lab with largest number of publications is chosen as 'parent' to

---

[6]In reality, the probability of a reviewer discovering a false positive may not be identical to that of discovering a false negative. Our symmetrical assumption here is one of simplicity.

reproduce. This algorithm strongly weights selection in favour of highly productive labs, which we view as an unfortunate but realistic representation of much of academic science. In the electronic supplementary material, we also report simulations using a weaker selection algorithm, for which all extant labs could be chosen as parent with a probability proportional to their number of published papers. We show that the results are marginally less dramatic than those reported in the main text, but are otherwise qualitatively similar.

Once a parent is chosen, a new lab with an age of zero is created, imperfectly inheriting the rigour of its parent lab with mutation. Specifically, lab $j$ with parent lab $i$ will have a false positive rate equal to

$$\alpha_j = \alpha_i + N(0, \epsilon), \tag{3.1}$$

where $N$ is a normally distributed random variable with a mean of zero and a standard deviation of $\epsilon$. Mutated values are truncated to stay within the range [0, 1].

## 3.3. Grant-seeking

In this final stage, labs apply for grant funding. A group of $d$ labs are selected at random to apply for grant funding, and one grant of size $G$ is awarded to a lab from this sample. The funded lab is chosen according to one of three allocation strategies described in the previous section. Under the PH strategy, the lab with the most published papers is awarded funding. Under the MI strategy, the lab with the lowest $\alpha$ value is awarded funding. Under the RA strategy, a lab is chosen at random for funding.

Hybrid strategies are implemented as follows. Under the MS, funds are allocated using the MI strategy a proportion $X$ of the time and the RA strategy otherwise. Under the ML strategy, funds are awarded randomly to the pool of *qualified* applicants. Applicants are qualified if their false positive rate is not greater than a threshold, $A$, such that the case of $A = 1$ is equivalent to the pure RA strategy.

We will show that such hybrid strategies, which are more realistically implemented than either of their constituent pure strategies, are quite effective at keeping false discovery rates low. In the real world, grants vary in size, and many grants are awarded by various agencies. Our modelling simplifications help to elucidate the effects of these parameters that are otherwise obscured by the heterogeneity present in real-world systems.

## 3.4. Computational experiments

We measure methodological rigour in scientific culture through the mean false positive rate of the scientific community (i.e. over all labs), $\bar{\alpha}$. We also record the total number of publications and the number of publications that are false discoveries (i.e. the results that do not match the correct epistemic state of the hypotheses), and by dividing the latter by the former, we can calculate the overall false discovery rate of the published literature, $F$. Both false positives *and* false negatives contribute to the false discovery rate. Note that the average false positive rate is an aggregate property of the labs performing scientific research, while the false discovery rate is a property of the published scientific literature.

We ran experiments consisting of 50 model runs for each set of parameter values tested (table 1). Each simulation was run for $10^7$ iterations to ensure convergence to a stable $\bar{\alpha}$, though most runs converged much more quickly, on the order of $10^5$ iterations. An iteration is not presumed to represent any specific length of time—our purpose is instead to illustrate more generally how selection works under our model's assumptions. Our model was coded in the D programming language [69]. The simulation code is available at https://github.com/mt-digital/badscience-solutions.

# 4. Results

Although our model is a dramatic simplification of how scientific communities actually work, it is still fairly complicated. We, therefore, take a piecemeal approach to our analysis so that the model dynamics can be more readily understood.

## 4.1. Comparing pure funding strategies in the absence of open science improvements

We first observed the impact of three pure funding strategies (PH, RA and MI) in the absence of open science improvements ($p = 0$, $r = 0$). This absence may be seen by some as an extreme condition, but

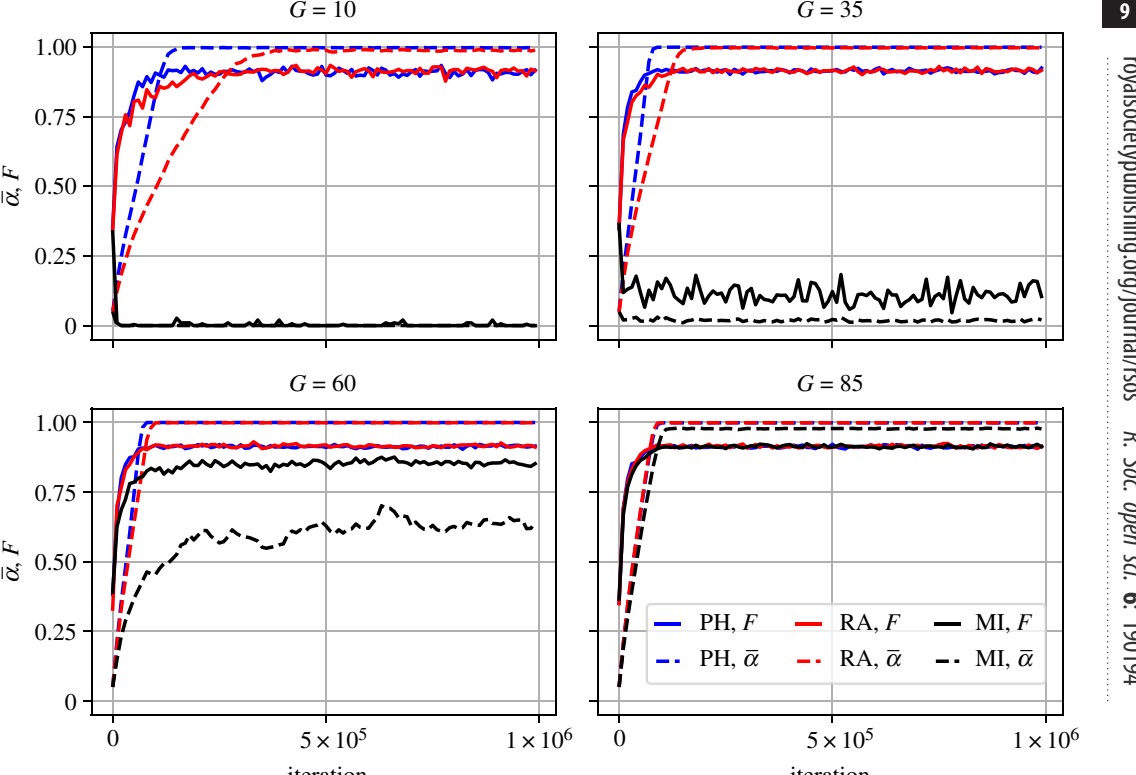

**Figure 2.** False positive rate ($\bar{\alpha}$, dashed lines) and false discovery rate ($F$, solid lines) over $10^6$ iterations for all three funding strategies (PH, RA and MI) across several grant sizes, $G$. $p = 0$, $r = 0$.

it serves as an valuable baseline. We find that funding based on PH leads to runaway increase of the false discovery rate (figure 2). This is unsurprising, as this funding strategy simply reinforces the selection pressure for publications that led to the degradation of methods in the analyses of Smaldino & McElreath [1]. Notably, RA of funds slows down the dynamic, but the situation in the long run is no better than when allocating funds based on publication history. In the electronic supplementary material, we show the differences between these two funding strategies to be negligible across wide variety of conditions.

MI, on the other hand, does an excellent job in keeping the false discovery and false positive rates low, particularly when the size of grants ($G$) is small (and therefore when scientists must receive many grants throughout their careers to remain productive). We consider small $G$ to represent a realistic scenario in most empirical fields. However, we note that if individual grants are very large, early success matters more. Whether an early career researcher receives a grant is largely stochastic, and long-term success is based on maximizing publications at any cost. Any competitive advantage among labs who are funded early in their careers regarding their rates of publication will be positively selected for. Thus, when grants are very large, even a funding strategy that only funds the most rigorous research can be associated with the eventual degradation of methods. Larger $G$ may, therefore, better reflect the case in which early successes cascade into a 'rich get richer' scenario [63].

We also find that the MI funding strategy decreases the total number of publications in the literature relative to the PH and RA strategies (electronic supplementary material). This occurs because only the labs using very rigorous methods are able to secure funding and therefore to publish continuously. These labs are less likely to produce false positives but also produce fewer total publications when there is a bias toward publishing positive results and as long as the base rate $b$ is less than 0.5 (a condition we believe is usually met). Thus, a funding strategy focused on MI may lead to less research being published. Whether or not this is a good thing for the advancement of scientific knowledge is open to debate.

## 4.2. Publishing negative results reduces false discovery, but only if negative results are equivalent to positive results

Next, we explore increasing the rate of publishing negative results (figure 3). We find that publishing negative results can decrease the false discovery and false positive rates, but, at least under PH and

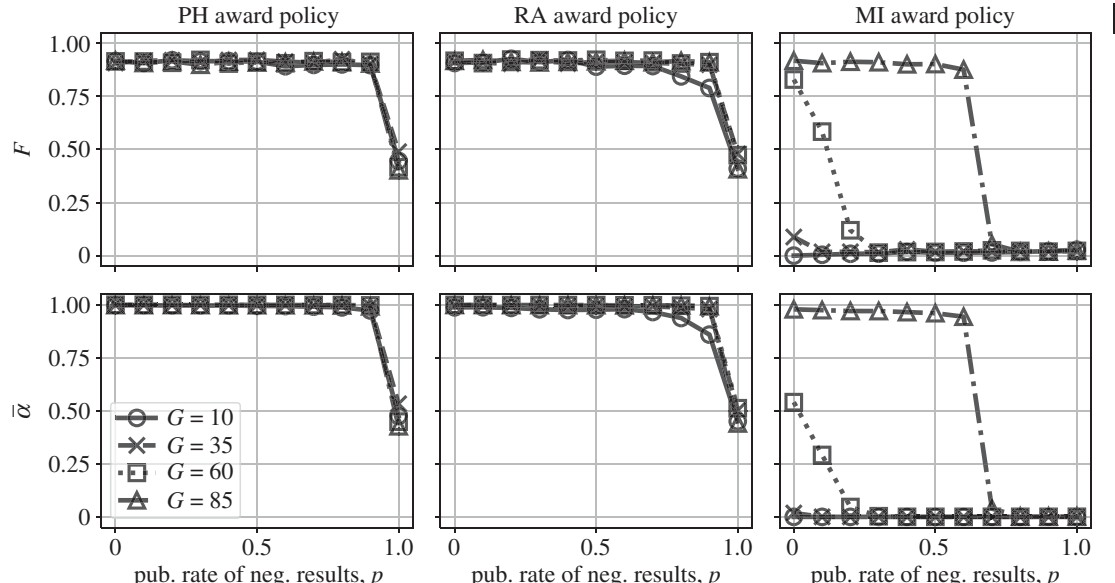

**Figure 3.** False discovery rate ($F$) and false positive rate ($\bar{\alpha}$) when negative results are published with varying frequency ($p \geq 0$, $r = 0$).

RA funding strategies, only when negative results are published at a similar rate as positive results (or, equivalently, only when negative results are equal or nearly equal in prestige to positive results). When the rate of publishing negative results is very high, RA slightly outperforms the PH strategy, as seen in figure 3. Only when $p = 1$ and publication bias is completely eliminated can labs with more rigorous methods effectively compete with those that can more readily obtain false positives.

With funding allocation based on MI, publishing negative results at even low rates can mitigate the early advantages from large grant amounts ($G$) described above. This is because the ability to profitably publish negative results removes some of the advantage that lower rigour engenders. Conversely, reducing publication bias without any additional incentives for rigour is, perhaps counterintuitively, unlikely to reduce the rate of false discovery in the scientific literature.

## 4.3. Improving peer review reduces false discovery, but only if reviewers are very effective

Here, we examine what happens when peer reviewers act as effective filters for erroneous results. Erroneous results are blocked from publication with probability $r$. Under the PH and RA funding strategies, we find that effective peer review helps reduce false discovery only when it is nearly perfect (figure 4). It is noteworthy that for very effective—but *not* perfect—peer review, we find a decrease in the false discovery rate (the proportion of false findings in the published literature) but not in the average false positive rate of the individual labs. That is, there is a mitigation of the natural selection of bad science, but not the natural selection of bad *scientists*. Instead, peer review acts as a filter to improve the published literature even when science is filled with bad actors. In reality, it is rather unlikely that peer review could improve so dramatically while the same scientists who review are also producing such shoddy work. In the presence of strong publication bias for positive results, publishing is still a numbers game: those who submit more get published more.

Under the MI funding strategy, we find that even a small improvement to peer review helps to lower both false discoveries in the literature and labs' false positive rates, and that this is true even for large $G$. This is because effective peer review reduces some of the advantage to those who have early successes but have high false positive rates. As with publication bias, improving peer review without any additional incentives for rigour is unlikely to substantively reduce the rate of false discovery.

## 4.4. The effects of publishing negative results and improving peer review interact

When it comes to lowering the false discovery rate in the published literature, the effects of publishing negative results and improving peer review can work in concert. For any level of peer review quality

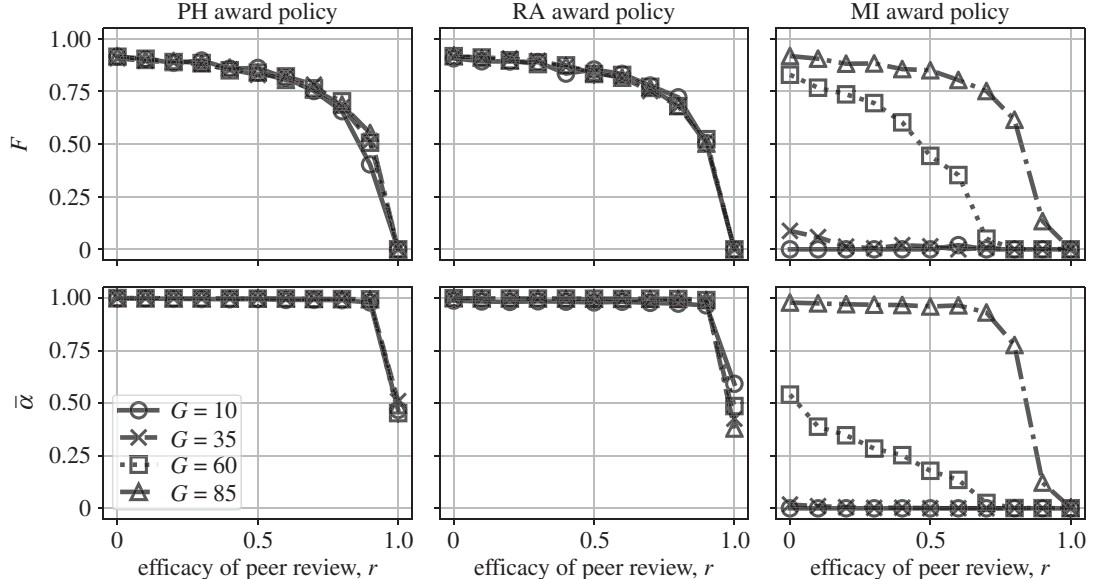

**Figure 4.** False discovery rate ($F$) and false positive rate ($\bar{\alpha}$) under improved peer review ($r \geq 0$, $p = 0$).

($r$), increasing the publication and prestige of negative results ($p$) will also lower the false discovery rate relative to baseline, with the exception of the (unlikely) scenario where peer review is perfectly accurate due to floor effects. Similarly, for any level of publishing negative results, improving the quality of peer review always lowers the false discovery rate (figure 5*a*,*b*).

For almost every scenario, however, the improvements to the published literature are much more substantial than the improvements to the scientists performing the research. That is, the average false positive rates of the individual labs stay high for most parameter values (figure 5*c*,*d*). Thus, in the absence of explicit rewards for rigour (e.g. in the form of grant funds), open science improvements may not be sufficient to improve science in the long run. They do not improve the scientific research being *performed*, but only the research that ends up being *published*. This is because when positive results have even a small advantage, when peer review is imperfect, and when selection ultimately favours productivity, those methods which allow researchers to maximize their publishable output will propagate. When funding agencies exclusively target those researchers using the most rigorous methods (figure 5, right column), however, open science improvements *can* interact to make a substantial difference in the type of scientific practices that are incentivized.

## 4.5. Hybrid funding strategies are effective at reducing false discoveries

The results presented so far paint a bleak picture. Open science improvements, by themselves, do little to reduce false discoveries at the population level. Removing selection for prestige at the funding stage does little as well. Only a concerted focus on methodological rigour—awarding funds to the most rigorous labs—seems to make much of a difference, and the feasibility of such an approach is dubious. This raises the question, however, of just how *much* of a focus on rigour is actually needed to reduce false discoveries. We tackle this question using two variations that combine RA with some focus on MI. Based on our finding that the effects of publishing negative results and improved peer review were essentially additive (figure 5), we restrict our analyses here to the case where $p = r$, reflecting the general extent of open science improvements.

We first consider the simple MS. A proportion $X$ of the time, funds are allocated to the lab with the lowest false positive rate, as in the MI funding strategy. The other $1 - X$ of the time, funds are awarded randomly as in the RA strategy. We find that when grants ($G$) are small, even a small percentage of funding going to the most rigorous labs has a very large effect on keeping the false discovery rate low, and this effect is aided by even small improvements to peer review and publication bias. As the size and importance of individual grants increases, larger improvements to $p$ and $r$ are required, but notably these improvements are still substantially smaller than what is required under the previously considered funding models. When grants are very large and a single grant early in a researcher's

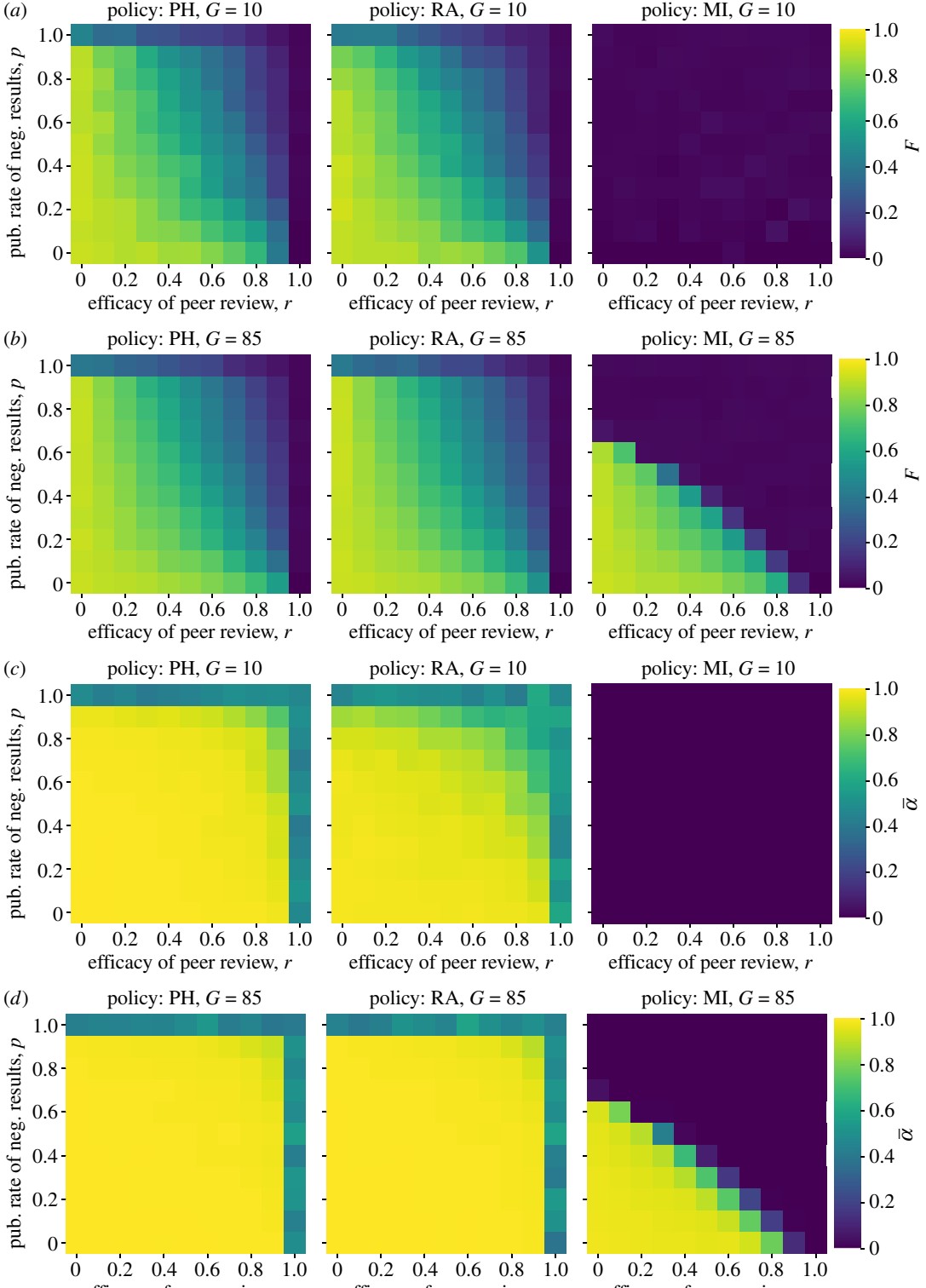

**Figure 5.** Reducing publication bias and improving peer review can work together to improve the quality of published research. (*a*) False discovery rate (*F*) with varying publication parameters for *G* = 10; (*b*) false discovery rate (*F*) with varying publication parameters for *G* = 85; (*c*) false positive rate ($\bar{\alpha}$) with varying publication parameters for *G* = 10 and (*d*) false positive rate ($\bar{\alpha}$) with varying publication parameters for *G* = 85.

career can therefore signify substantial advantages, larger improvements to *p* and *r* are necessary to keep false discoveries low (figure 6).

Next, we consider allocating funds using an ML. This strategy is most similar to what has recently been proposed by funding reform advocates. Funds are awarded randomly to the pool of *qualified*

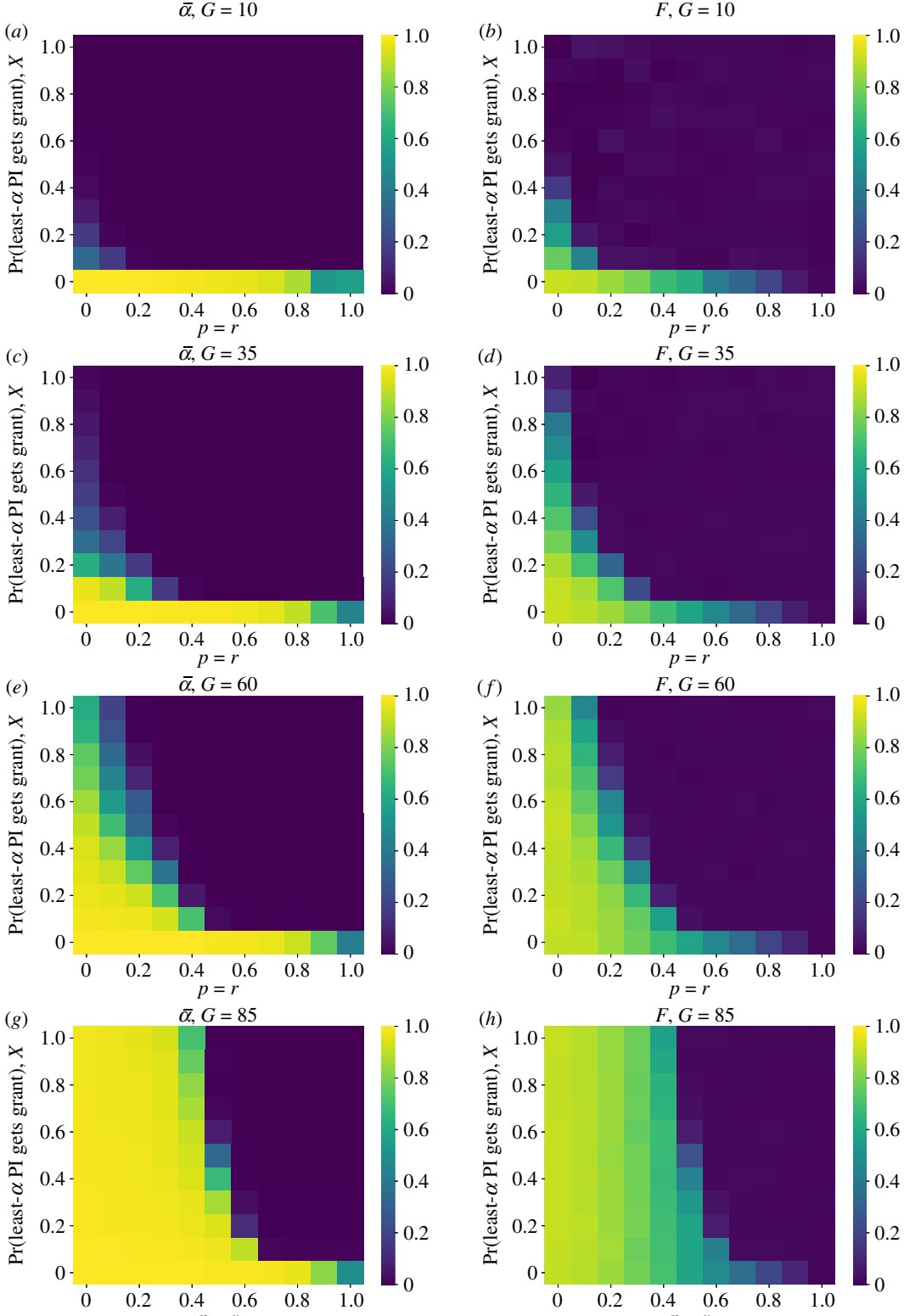

**Figure 6.** Average false positive rate ($\bar{\alpha}$) and false discovery rates ($F$) under mixed strategy (MS) funding allocation for varying rates of funding rigour ($X$), open science improvements ($p = r$) and funding level ($G$). (a) False positive rate, $G = 10$; (b) false discovery rate, $G = 10$; (c) false positive rate, $G = 35$; (d) false discovery rate, $G = 35$; (e) false positive rate, $G = 60$; (f) false discovery rate, $G = 60$; (g) false positive rate, $G = 85$ and (h) false discovery rate, $G = 85$.

applicants. Applicants are qualified if their false positive rate is not greater than a threshold, $A$. We find that this strategy can be quite effective at keeping the false discovery rate low. Importantly, the threshold, $A$, can be fairly high. Even the case where labs with false positive rates of up to 20 or 30 per cent are

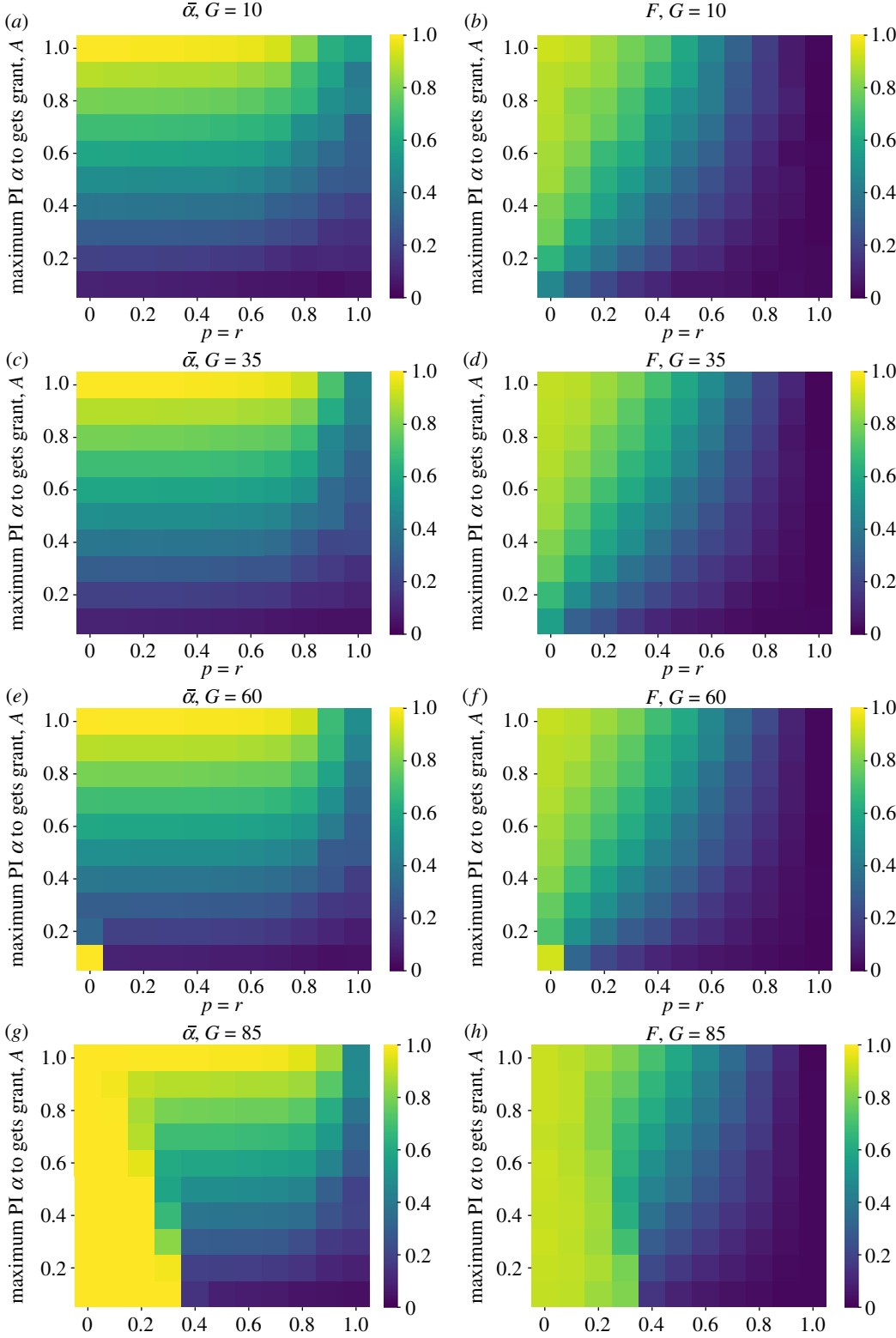

**Figure 7.** Average false positive rate ($\bar{\alpha}$) and false discovery rates ($F$) under the *modified lottery* (ML) funding strategy for varying rigour threshold ($A$), open science improvements ($p = r$) and funding level ($G$). (a) False positive rate, $G = 10$; (b) false discovery rate, $G = 10$; (c) false positive rate, $G = 35$; (d) false discovery rate, $G = 35$; (e) false positive rate, $G = 60$; (f) false discovery rate, $G = 60$; (g) false positive rate, $G = 85$ and (h) false discovery rate, $G = 85$.

entered into the lottery still produced a marked reduction in the false discovery rate. If grants ($G$) are very large (and so initial success plays an outsized influence on overall success), then the ML must be compensated by increased contributions from open science improvements (figure 7).

# 5. Discussion

Under a model of career advancement that makes publication quantity paramount to hiring and promotion, can journals, academic societies and funding agencies nevertheless implement changes to mitigate the natural selection of bad science? Our results suggest a cautious affirmative. However, such changes are not trivial, and will garner the best results when they are implemented in tandem.

Randomly allocating research funds, as with a lottery system, may confer several advantages over a system favouring publishing history or related factors, such as prestige or the 'hotness' of a topic [18–22]. Lotteries may reduce gender or institutional bias in the allocation of funding, and facilitate more effective use of researchers' time, which can ultimately lead to more science being done. However, lotteries are inherently neutral and therefore cannot oppose strong selective forces. Any advantage for more replicable science will come not from a random component but from a directed emphasis on methodological rigour. Our model indicates that a pure funding strategy of RA will produce nearly identical results as a funding strategy favouring highly productive researchers.

Funding strategies that specifically target methodological rigour, on the other hand, can have very important consequences for the future of science, even in the face of career incentives for publication at the levels of hiring and promotion. Two aspects of this result allow some room for optimism. First, funders' focus need not be entirely dedicated to rigour. If even a relatively small proportion, say 20 per cent, of grants were dedicated to the most rigorous proposals, science as a community would benefit. Some caveats apply. Our results assume that the remaining grants are allocated at random. Nevertheless, our analyses suggest that a funding strategy that specifically targets publication history is little worse than a purely random funding allocation strategy. A more serious caveat is that rigour is notoriously difficult to infer, and any such inference may be costly in terms of the person-hours required to make such an assessment. Automated assessments risk being gamed, as all algorithms for social decision-making do [70]. A second aspect of our results offers a potential solution. Our study of modified lotteries indicates that the threshold for rigour does not need to be unrealistically high to yield important benefits. For example, under the parameters we explored, a lottery that excluded only those labs with an average false positive rate of 30% or higher would, in many cases produce a 60% reduction in the false discovery rate relative to a pure lottery or publication-based allocation strategy. Moreover, this improvement will only get better as open science improvements yield more widespread effects.

Funders are, of course, interested in more than rigour. The most rigorous science, defined in our model as the least likely to yield false positives, may also be desperately uninteresting. Interesting science teaches us something new about our universe, and therefore often involves uncertainty at the outset. Important science also serves a function that allows us to change our world for the better. For these reasons, funders are also interested in innovation and application. It is at present unclear how rewards for rigour will or should interact with rewards for novelty or applied research. Research that is path-breaking but cuts corners might compete with research that is rigorous but trivial. Exactly how this interaction between rigour, novelty, and applicability plays out is an important focus for future research.

Our model assumes that all research requires funding. In reality, some research requires little or no funding. Other research may be funded by sources driven more by novelty, prestige, or charisma. As such, a PI who pursues funding driven by MI may suffer, because they must sacrifice some degree of productivity or novelty. On the other hand, if sufficient prestige becomes associated with such rigour-based funding, the detrimental effects of fewer publications may be mitigated, yielding a kind of 'two paths' model of academic success. Such a model may indeed be a good representation of some modern academic disciplines.

Overall, our results indicate that funding agencies have the potential to play an outsized role in the improvement of science by promoting research that passes tests for rigour. Such tests include commitments to open data and open code (which permit closer scrutiny), preregistration and registered reports, and research programs with strong theoretical bases for their hypotheses. Wide-scale adoption of these and similar criteria for funding agencies can, in theory, have substantial long-term effects on reducing the rates of false discoveries.

Our results also highlight the contribution of open science practices. Improving peer review and reducing publication bias led to improvements in the replicability of published findings in our simulations. Alone, each of these open science improvements required extremely high levels of implementation to be effective. Fortunately, we also found that the two factors could work in tandem to improve the replicability of the published literature at lower, though still high, levels of efficacy. Unfortunately, in the absence of specific incentives at the funding or hiring level for methodological

rigour, open science improvements are probably not sufficient to stop the widespread propagation of inferior research methods, despite the optimism that often surrounds their adoption. Moreover, it is not unreasonable to harbour doubts about the extent to which policies that improve methods will become mainstream in a system that nevertheless rewards those who cut corners. When combined with funding strategies that explicitly promote rigour, however, open science improvements can make powerful contributions to more reproducible science.

Rapid institutional changes that incentivize the publication or prestige of negative results, including failed replications, and improve the quality of peer review may end up having a relatively small effect on the long-term reproducibility of science, but that does not make them unimportant. As we see in our model, even in the absence of any incentives for rigour at the funding or hiring level, such changes can interact to improve the quality of the published literature. Such changes should therefore be encouraged. Moreover, there are likely benefits to such changes that are not included in our model, beyond the immediate reduction of false discoveries [24,71]. They may create a more transparent system of science that improves quality and provides better training for future scientists. They may help improve future research by promoting a more accurate literature today, because researchers build on previous publications—in reality, hypotheses tested in different cycles are not fully independent. They may help to mitigate pernicious biases based on gender, race and geography. They may create new markers of prestige that actively incentivize best practices. And they may help to create a culture of accountability and verifiability, allowing science to better live up to the Royal Society's motto *Nullius in verba*. Solving complicated problems like the ones facing academic science requires creating common knowledge [72]. It is only after we all understand what the problems are and what solutions might look like that working together toward a collective solution becomes possible.

Even if a community of researchers agree on the superiority of certain methods or approaches, and even if there is no penalty in terms of publishing metrics to their use, there is still no guarantee that those methods or approaches will be widely adopted. Currently, few funders use lotteries. Measuring the adoption of open science practices is not straightforward, but in most fields, it is still the case that few published studies are preregistered. Most journals do not require open data and code, and even among those that do there is no guarantee that such data and code are usable to reproduce the paper's analyses [50]. What influences adoption of best practices? In a well-known theoretical study, Boyd & Richerson [73] showed that group-beneficial norms are most likely to spread when the associated benefit is large and apparent, and when individuals using different norms interact regularly so that those using the inferior norm can observe the benefits of switching. These findings imply that tracking the success of open science norms and the impact of new funding strategies is imperative, as is promoting those successes. As an example, McKiernan *et al.* [74] make the case that research papers reporting open science practices receive more citations and media coverage than comparable papers that do not use those practices.

That said, proponents of open science should avoid gloating. Also imperative is that individuals who promote open science interact often and respectfully with non-converts. For one thing, skeptics often have valid concerns. It may be all too easy to adopt the veneer of open science practices without internalizing deeper concerns for rigour and thoroughness. If the signals of open science end up being rewarded without requiring the commitments those signals are intended to convey, then we are back to square one, just as publication quantity and journal impact factor do not align with our ideals of scientific productivity and influence. Moreover, scientists, like most humans, are group-ish. Akerlof & Michaillat [55] recently demonstrated how inferior paradigms can persist when paradigms are related to identities that incentivize the gatekeepers of science rewarding their own. In a rich treatment of this idea, Francisco Gil-White has referred to the phenomenon as 'paradigm rent seeking' (Gil-White, F. Academic market structure and the demarcation problem: Science, pseudoscience, and a possible slide between. Unpublished manuscript.) In such cases as well, unambiguous and consistent demonstrations of the superiority of better methods and practices are paramount in ensuring their adoption.

In our analysis, we found a wide range of conditions under which the false discovery rate of publications fell much more than the average false positive rate of individual labs. It appears that some institutional changes can effectively reduce the number of false discoveries from ending up in the published literature but simultaneously fail to improve the overall quality of the scientists who produce those discoveries. A small contribution to this effect arises from the fact that regardless how high the false positive rate is, some findings will still be correct. However, the effect is primarily driven by the coexistence of strong levelling mechanisms (reducing publication bias and improving peer review) that reduce variability in journal publications, along with strong selection mechanisms at the hiring and promotion bottlenecks that continue to favour individuals who can nevertheless get more papers

published. If this situation reflects the current or emerging landscape of open science and academic incentives, it should cause us some concern. Formal institutions made of rules and regulations—like at least some of the incentives for open science improvements—are top-down constraints, and as such can be changed fairly rapidly [23]. More deeply ingrained norms of conduct—like the methods and paradigms that shape how science is produced in the lab—involve tacit knowledge and internalized associations that are far less malleable [75–77]. If our incentives are not powerful enough to change those norms over time via cultural evolution, then our scientific communities remain in peril from any shocks that might disrupt the institutions promoting best practices. Such a shock could lead the system to rapidly revert to publishing low-quality science at high rates. Preventing this kind of system-wide fragility requires either changing the fundamental incentives of academic science (e.g. not rewarding behaviours associated with high rates of false positives) or introducing countervailing selection pressures (e.g. actively rewarding behaviours associated with low rates of false positives).

Our model obviously reflects a highly simplified view of science. In particular, we focus on a view of science as the accumulation of facts. Our model is utility-maximizing under the assumption that higher utility always comes from greater accumulation of more facts known with increasing certainty. Facts are indeed the raw ingredients of science, but the meal does not get made without proper theory to organize those facts. Moreover, as philosophers of science have noted, scientific theories are embedded in scientists' worldviews [78], and must either be assimilated into the beliefs, norms, and goals of those scientists or else force those beliefs, norms, and goals into better accordance with those theories. A complementary approach to ours, then, is to consider alternative utility functions to describe an ideal picture of science, and consequently how institutional forces might shape the cultural evolution of scientific practices in relation to those utilities.

In the short run, we encourage institutional efforts that increase the publication of negative results, enforce methodological rigour in peer review, and above all, attempt to funnel funding toward high-integrity research. In the long run, these changes are probably not sufficient to ensure that methodological and paradigmatic improvements are consistently adopted. Ultimately, we still need to work toward institutional change at those great bottlenecks of hiring and promotion. We should strive to reward *good* science that is performed with integrity, thoroughness, and a commitment to truth over what is too often seen as 'good' science, characterized by flawed metrics such as publication quantity, impact factor and press coverage.

Ethics. Upon completing the experiment, all simulated scientists transcended this mortal realm to reside forever in digital nirvana.
Data accessibility. Only simulated data were used for our analyses. Model code is made available at https://github.com/mt-digital/badscience-solutions.
Authors' contributions. P.E.S. conceived the project. P.E.S., P.C.K. and M.A.T. designed the model. M.A.T. coded and analysed the model. P.E.S. wrote the paper. All authors edited and reviewed the paper.
Competing interests. We have no competing interests.
Funding. Computational experiments were performed on the MERCED computing cluster, which is supported by the National Science Foundation (grant no. ACI-1429783). This work was funded by DARPA grant no. HR00111720063 to P.E.S. The views and conclusions contained herein are those of the authors and do not necessarily represent the official policies or endorsements of DARPA or the USA Government.
Acknowledgements. This paper was made better thanks to helpful comments from John Bunce, Daniël Lakens, Karthik Panchanathan, Anne Scheel, Leo Tiokhin and Kelly Weinersmith.

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
