## [Reviewer comments · Royal Society Open Science]

Review History

RSOS-190194.R0 (Original submission)

Review form: Reviewer 1

Is the manuscript scientifically sound in its present form?

Yes

Are the interpretations and conclusions justified by the results?

Yes

Is the language acceptable?

Yes

Is it clear how to access all supporting data?

Yes

Do you have any ethical concerns with this paper?

No

Have you any concerns about statistical analyses in this paper?

No

Recommendation?

Accept as is

Comments to the Author(s)

This study builds from previous work to show that practices that reward quantity instead of quality of scientific publishing risk promoting bad science. The efficacy of science as a collective endeavor then suffers, both in the sense of making the published literature less reliable, and in the sense of rewarding scientists whose practices are more apt to generate erroneous results. These results help build the case for adopting open science practices and valuing negative results.

At an operational level, the manuscript does everything that a good modeling study should do. The model and its assumptions are clearly articulated, an interesting question is posed, and the question is explored thoroughly and honestly. Limitations of the model are discussed plainly, and interesting subsequent lines of investigation proposed. All of the ingredients that one expects are here, and the reader will have no trouble parsing the manuscript and understanding the essential results. The manuscript is also written with a satisfying stylistic flair that is rare among scientific articles.

The relevance of the model hinges on whether the assumptions on which it rests faithfully reflect the actual state of affairs in science. To be sure, science is so vast that any attempt to capture some aspect of how the process works will invariably require strong assumptions. In these early days of science as a focus of inquiry itself, it is healthy to have a variety of models of science in the literature for readers and investigators to contemplate. Whether the assumptions on which this particular model is based are justified or accurate is fodder for debate, but such debate is healthy and drives the field forward. While I don't agree with every assumption here, a case could certainly be made in support of all the major assumptions on which this model rests, and that is all that one can reasonably ask.

For what it's worth, I would argue for a base rate higher than 10%, but I appreciate that the additional simulations in the Supplement demonstrate that the key results are robust to base rates across a large range. Also, though it is not central to the manuscript's essential findings, I wouldn't agree that the primary role of peer review to weed out erroneous conclusions. One can hope that peer review will weed out the most egregiously flawed work, but most flawed papers are not so obviously wrong that a dedicated and conscientious reviewer could, by sheer strength of intellect, notice. Many of the practices discussed here under the banner of effective peer review (registered reports, double-blind review, improved statistical training) are entirely worthwhile, so perhaps my objection lies with the shorthand of labeling this collection of practices as "effective peer review". Nevertheless, I fear that the role of peer-review is often misunderstood, so would hesitate to further the notion that the primary role of peer review is to weed out erroneous results.

These are small quibbles, though. Overall, the manuscript clearly passes the standard for publication in this journal in its current form. I congratulate the authors on a nice contribution.

Review form: Reviewer 2 (Olmo van den Akker)

Is the manuscript scientifically sound in its present form?

Yes

Are the interpretations and conclusions justified by the results?

Yes

Is the language acceptable?

Yes

Is it clear how to access all supporting data?

Yes

Do you have any ethical concerns with this paper?

No

Have you any concerns about statistical analyses in this paper?

No

Recommendation?

Accept with minor revision (please list in comments)

Comments to the Author(s)

In this paper, the authors use computational modeling to assess whether open science improvements and funding lotteries are likely to improve the reproducibility of science. More specifically, they investigate the influence of three key factors: the publication of negative results, improved peer review, and the criteria for funding allocation.

I think that the authors make an important contribution by extending the 2016 model of the natural selection of bad science to include funding allocation and open science improvements. The inclusion of funding allocation is important because the 2016 model assumed that the rate at which research groups could produce results is limited only by the rigor of their methods. In my view, this assumption does not map well onto scientific reality and the availability of funding is a much more realistic way to model the potential output of research labs. The inclusion of open science improvements is important in the light of the increasing adoption of open science initiatives like preprints, open science badges, and registered reports. Although it makes intuitive sense that these improvements help us get to a literature with less false research findings, studies assessing the efficacy of these initiatives are currently lacking. While the current paper does not provide empirical evidence towards the efficacy of these initiatives, it does provide a model that makes valuable suggestions about which of the initiatives would be most worth pursuing. In all, extending the 2016 model this way has important benefits.

Even though I think the paper has enough value to be published as it is, I do have some comments and suggestions that may improve the quality of the paper. I have attached the points to this message (Appendix A). I discuss them in order, but I must emphasize that I think the points related to the model itself are most important.

If the authors have any questions or comments regarding my peer review, please do not hesitate to contact me.

With kind regards,

Olmo van den Akker (ovdakker@gmail.com)

Decision letter (RSOS-190194.R0)

24-Apr-2019

Dear Dr Smaldino

On behalf of the Editors, I am pleased to inform you that your Manuscript RSOS-190194 entitled "Open science and modified funding lotteries can impede the natural selection of bad science" has been accepted for publication in Royal Society Open Science subject to minor revision in accordance with the referee suggestions. Please find the referees' comments at the end of this email.

The reviewers and handling editors have recommended publication, but also suggest some minor revisions to your manuscript. Therefore, I invite you to respond to the comments and revise your manuscript.

- Ethics statement

- Data accessibility

<http://datadryad.org/submit?journalID=RSOS&manu=RSOS-190194>

- Competing interests

- Authors' contributions

- Acknowledgements

- Funding statement

Because the schedule for publication is very tight, it is a condition of publication that you submit the revised version of your manuscript before 03-May-2019. Please note that the revision deadline will expire at 00.00am on this date. If you do not think you will be able to meet this date please let me know immediately.

- 1) A text file of the manuscript (tex, txt, rtf, docx or doc), references, tables (including captions) and figure captions. Do not upload a PDF as your "Main Document";
- 2) A separate electronic file of each figure (EPS or print-quality PDF preferred (either format should be produced directly from original creation package), or original software format);
- 3) Included a 100 word media summary of your paper when requested at submission. Please ensure you have entered correct contact details (email, institution and telephone) in your user account;
- 4) Included the raw data to support the claims made in your paper. You can either include your data as electronic supplementary material or upload to a repository and include the relevant doi

within your manuscript. Make sure it is clear in your data accessibility statement how the data can be accessed;

5) All supplementary materials accompanying an accepted article will be treated as in their final form. Note that the Royal Society will neither edit nor typeset supplementary material and it will be hosted as provided. Please ensure that the supplementary material includes the paper details where possible (authors, article title, journal name).

on behalf of Professor Zoltan Dienes (Associate Editor) and Essi Viding (Subject Editor)
openscience@royalsociety.org

Associate Editor Comments to Author (Professor Zoltan Dienes):

Associate Editor: 1

Comments to the Author:

I agree with the reviewers that your paper makes a clear and interesting contribution; I think addressing the impact of different funding schemes of scientific integrity is also timely. I think the manuscript stands clearly as it is; nonetheless, the reviewers provide thoughtful comments to take on board for a revision.

Reviewer comments to Author:

Reviewer: 1

Comments to the Author(s)

This study builds from previous work to show that practices that reward quantity instead of quality of scientific publishing risk promoting bad science. The efficacy of science as a collective endeavor then suffers, both in the sense of making the published literature less reliable, and in the sense of rewarding scientists whose practices are more apt to generate erroneous results. These results help build the case for adopting open science practices and valuing negative results.

At an operational level, the manuscript does everything that a good modeling study should do. The model and its assumptions are clearly articulated, an interesting question is posed, and the question is explored thoroughly and honestly. Limitations of the model are discussed plainly, and interesting subsequent lines of investigation proposed. All of the ingredients that one expects are here, and the reader will have no trouble parsing the manuscript and understanding the essential results. The manuscript is also written with a satisfying stylistic flair that is rare among scientific articles.

The relevance of the model hinges on whether the assumptions on which it rests faithfully reflect the actual state of affairs in science. To be sure, science is so vast that any attempt to capture some aspect of how the process works will invariably require strong assumptions. In these early days of science as a focus of inquiry itself, it is healthy to have a variety of models of science in the literature for readers and investigators to contemplate. Whether the assumptions on which this particular model is based are justified or accurate is fodder for debate, but such debate is healthy and drives the field forward. While I don't agree with every assumption here, a case could certainly be made in support of all the major assumptions on which this model rests, and that is all that one can reasonably ask.

For what it's worth, I would argue for a base rate higher than 10%, but I appreciate that the additional simulations in the Supplement demonstrate that the key results are robust to base rates across a large range. Also, though it is not central to the manuscript's essential findings, I wouldn't agree that the primary role of peer review is to weed out erroneous conclusions. One can hope that peer review will weed out the most egregiously flawed work, but most flawed papers are not so obviously wrong that a dedicated and conscientious reviewer could, by sheer strength of intellect, notice. Many of the practices discussed here under the banner of effective peer review (registered reports, double-blind review, improved statistical training) are entirely worthwhile, so perhaps my objection lies with the shorthand of labeling this collection of practices as "effective peer review". Nevertheless, I fear that the role of peer-review is often misunderstood, so would hesitate to further the notion that the primary role of peer review is to weed out erroneous results.

These are small quibbles, though. Overall, the manuscript clearly passes the standard for publication in this journal in its current form. I congratulate the authors on a nice contribution.

Reviewer: 2

Comments to the Author(s)

In this paper, the authors use computational modeling to assess whether open science improvements and funding lotteries are likely to improve the reproducibility of science. More specifically, they investigate the influence of three key factors: the publication of negative results, improved peer review, and the criteria for funding allocation.

I think that the authors make an important contribution by extending the 2016 model of the natural selection of bad science to include funding allocation and open science improvements. The inclusion of funding allocation is important because the 2016 model assumed that the rate at which research groups could produce results is limited only by the rigor of their methods. In my view, this assumption does not map well onto scientific reality and the availability of funding is a much more realistic way to model the potential output of research labs. The inclusion of open science improvements is important in the light of the increasing adoption of open science initiatives like preprints, open science badges, and registered reports. Although it makes intuitive sense that these improvements help us get to a literature with less false research findings, studies assessing the efficacy of these initiatives are currently lacking. While the current paper does not provide empirical evidence towards the efficacy of these initiatives, it does provide a model that makes valuable suggestions about which of the initiatives would be most worth pursuing. In all, extending the 2016 model this way has important benefits.

Even though I think the paper has enough value to be published as it is, I do have some comments and suggestions that may improve the quality of the paper. I have attached the points to this message. I discuss them in order, but I must emphasize that I think the points related to the model itself are most important.

If the authors have any questions or comments regarding my peer review, please do not hesitate to contact me.

With kind regards,
Olmo van den Akker (ovdakker@gmail.com)

Author's Response to Decision Letter for (RSOS-190194.R0)

See Appendix B.

Decision letter (RSOS-190194.R1)

04-Jun-2019

Dear Dr Smaldino,

I am pleased to inform you that your manuscript entitled "Open science and modified funding lotteries can impede the natural selection of bad science" is now accepted for publication in Royal Society Open Science.

Royal Society Open Science operates under a continuous publication model (<http://bit.ly/cpFAQ>). Your article will be published straight into the next open issue and this

will be the final version of the paper. As such, it can be cited immediately by other researchers. As the issue version of your paper will be the only version to be published I would advise you to check your proofs thoroughly as changes cannot be made once the paper is published.

on behalf of Professor Zoltan Dienes (Associate Editor) and Essi Viding (Subject Editor)
openscience@royalsociety.org

Associate Editor Comments to Author (Professor Zoltan Dienes):

You have addressed the reviewers' points well, and your paper can now be accepted. I hope it has the impact it deserves in changing how funding is allocated!

Appendix A

Open science and modified funding lotteries can impede the natural selection of bad science

By: Paul Smaldino, Matthew Turner, & Pablo Contreras Kallens

In this paper, the authors use computational modeling to assess whether open science improvements and funding lotteries are likely to improve the reproducibility of science. More specifically, they investigate the influence of three key factors: the publication of negative results, improved peer review, and the criteria for funding allocation.

I think that the authors make an important contribution by extending the 2016 model of the natural selection of bad science to include funding allocation and open science improvements. The inclusion of funding allocation is important because the 2016 model assumed that the rate at which research groups could produce results is limited only by the rigor of their methods. In my view, this assumption does not map well onto scientific reality and the availability of funding is a much more realistic way to model the potential output of research labs. The inclusion of open science improvements is important in the light of the increasing adoption of open science initiatives like preprints, open science badges, and registered reports. Although it makes intuitive sense that these improvements help us get to a literature with less false research findings, studies assessing the efficacy of these initiatives are currently lacking. While the current paper does not provide empirical evidence towards the efficacy of these initiatives, it does provide a model that makes valuable suggestions about which of the initiatives would be most worth pursuing. In all, extending the 2016 model this way has important benefits.

Even though I think the paper has enough value to be published as it is, I do have some comments and suggestions that may improve the quality of the paper. I discuss the points in order, but I must emphasize that I think the points related to the model itself are most important.

Points related to the Introduction:

- 1) On page 5 the authors state that open science developments and changes to funding schemes can occur rapidly. Technically this is true as institutions can quickly change their policies. However, whether they will do so depends largely on cultural norms. Because cultural change can take a while, I would recommend the authors to be more nuanced in their discussion of the pace with which these developments can come about. Rephrasing the term 'rapid institutional changes' can go a long way here.
- 2) On page 6 the authors point out that the publication of negative results is getting more popular. The authors only mention registered reports as evidence for this development, but they could also mention the increased popularity of large-scale replication projects like Many Labs and RP:P. In addition, the authors mention some journals that emphasize their willingness to publish null results on page 5, but I feel that that passage would be more suitable in this section.
- 3) On pages 6-9 the authors present several questions related to the extensions of the 2016 model. These questions appear to be the authors' main research questions. However, these questions are only implicitly answered in the results section. Explicitly restating and answering these questions would provide more structure to the paper.

Points related to the model itself:

- 1) On page 10 the authors mention two hybrid strategies they included in the model: MS and ML, which are combinations of MI and RA, and RA and a threshold value of the false positive rate

respectively. It makes sense to include such hybrids, but it is unclear why the authors only choose to implement these hybrids in the model. Why do the authors not incorporate a mix between PH and RA, and PH and MI, or even a mix of all three pure strategies? Some elaboration on the authors' choices would be welcome here.

- 2) On page 12, Figure 1 indicates that the order of the stages in the model is 1) Science, 2) Evolution, 3) Grant-seeking. However, in my opinion it would make more sense if Grant-seeking is the first stage because labs can only do science if they have previously been awarded grant money.
- 3) On page 12/13, the authors explain that a lab with enough funds will select a hypothesis to investigate in the Science phase. This hypothesis is true with a probability of b (the base rate of the field). In the model, b is an exogenous variable that does not change throughout the simulation. However, in reality, the types of hypotheses that labs choose may be highly dependent on the incentive structure in their field. For example, if novelty is the main funding criterion in the field, labs will look for hypotheses that are novel but unlikely to be true instead of hypotheses that are unimaginative but are likely to be true. As the model involves different incentive structures I think it would make sense to include b as an endogenous variable in the model.
- 4) On page 13 the authors state that they assume a power of $W = 0.80$ in their model, but studies show that empirical power is much lower, at least in psychology (Bakker et al., 2012). It would be interesting to see the results of the simulation with a power of 0.30 or 0.40.
- 5) On page 13 the authors note that in the 2016 model increased rigor not only yielded fewer false positives but also decreased the rate at which labs could produce new results and thereby submit new papers. In the current paper the authors disregard this assumption for reasons of tractability. Although I understand the need to avoid a model that is too complex, the authors do not provide a convincing argument why this assumption needed to be disregarded and not other parts of the model. An explanation and the presentation of results that do include this disregarded assumption would be welcome to assess the robustness of the findings.
- 6) On page 14, the authors explain that in their model peer review works to keep false positives and false negatives out of the published literature. However, in reality, reviewers do not have accurate knowledge about the true state of the world and should not focus on the results of a paper but on its methodological rigor. Of course, reviewers currently do focus on the results of a paper, but they tend to focus on whether the results are positive and not on whether the results correspond to reality. In my opinion, it makes more sense for peer review to be operationalized similarly to the MI funding scheme. That is, peer reviewers would accept a paper for publication when it exceeds a threshold value of methodological integrity. This operationalization would also alleviate the need for the dubious assumption that peer review is equally capable of weeding out false positives and false negatives.
- 7) On page 14, the authors explain that in the Evolution stage each iteration a lab dies out and a new lab is created that adopts the methods of a progenitor lab. This means that the total number of labs in the simulation remains constant, which might not be in line with the fact that almost all fields of science have been increasing in the last few decades. Adding more labs instead of one may account for the constantly increasing size of scientific output.
- 8) On page 14, the authors explain that in the Evolution stage the false positive rate of a newly created lab is determined by the false positive rate of their parent lab and a random mutation.

This mutation is truncated to only be positive, which might make sense given that new labs might not have as much methodological experience as older labs. However, the authors do not explain this choice, so the reader is left wondering.

- 9) On page 16, the authors state that each simulation was run for 10 million iterations to ensure convergence to a stable mean false positive rate. This number of iterations is very large and one can doubt whether this large number is applicable to the actual evolution of science. If we suppose that every month somewhere in the world a research lab perishes and one is created, the number of iterations in the model would amount to more than 800.000 years of scientific evolution. If we suppose that it occurs every day, the number of years would still be more than 27.000. These are unrealistic timescales given that science as we know it is at most several hundred years old. Therefore, the large number of iterations in the model raises the question whether conclusions drawn from this simulation can be applied to the state of science in 10 or 20 years, which seems to be a main goal of the authors. I would be curious to see the results if the authors presented them on a more realistic timescale.
- 10) On page 15, the authors explain that it is randomly decided which lab applies for funding. However, applying for funding is not a random process as labs who focus on research / publishing papers have less time to apply for grants. One way this non-randomness could be incorporated in the model would be to only let labs apply for a grant if they have (nearly) run out of grant money. In my view, this would match reality better than the current model.

Point related to the Results:

- 1) On pages 16-26, the authors explain in several sections what happens in the model when certain parameters are tweaked. However, the explanations for section 4.2 and 4.5 especially are not entirely clear to me. I think the Results section could benefit from a phase-by-phase description of the developments that are shown in the related figures.

Points related to the Discussion:

- 1) On page 28, the authors state that “Such a model may indeed be a good representation of some modern academic disciplines”, but they do not state which disciplines this would be and why such a model would represent them well.
- 2) Throughout the paper, the authors sometimes refer to replication and preregistration, but the authors do not specifically model replication, preregistration, or registered reports. Modeling these developments specifically would be welcome as they are very much at the forefront of current scientific infrastructure. Maybe the authors could indicate in the Discussion that a formal model including these developments could be a valuable research line.

Points related to language/clarity:

- 1) On page 4 the authors note that “the computational model presented by Smaldino & McElreath (2016) made several pessimistic—if realistic—assumptions about the ecosystem of academic science. We focus on two. First, it was assumed that publishing negative or null results is either difficult or, equivalently, confers little or no prestige. Second, it was assumed that publishing novel, positive results is always possible. In other words, the model ignores the corrective role of peer review or, equivalently, assumes it is ineffective.” To me, it is not directly clear how peer

review is related to the prestige associated with publishing positive and negative results. One or two additional sentences explaining this link would be welcome.

- 2) On page 15 the authors use the term 'false discovery rate' to mean false positive and false negative findings in the literature. This term may be confusing because 1) its similarity to the term false positive rate, and 2) the fact that discovery implies a positive result (i.e., in common parlance, one cannot discover the absence of an effect). One option to relabel these terms would be 'lab false positive rate' and 'false publication rate', but of course there are many options.
- 3) On page 23 the authors mention: "Based on our finding that the effects of publishing negative results and improved peer review were cumulative, we use $p = r$ for simplicity". To me it is unclear what the authors mean by cumulative in this sentence and why it would lead to the assumption of $p = r$. This assumption would imply that the publication rate for negative results is equal to the efficacy of peer review, which does not seem to be well-aligned with reality.
- 4) On page 13 and 14 the authors mention that several additional robustness checks lead to "qualitatively similar" results. However, the authors do not specify what they mean by this phrase. Given that robustness checks are crucial in this kind of computational modeling, a specification of the qualitative nature of the results is desirable.

In all, I think the author's manuscript adds value to the current literature and provides us insight in the role of funding schemes and open science practices in attaining a more reproducible science. I do think that the model employed in the manuscript could use some additional parameter clarifications and robustness checks, but if those are implemented I would be happy to see the manuscript published in Royal Society Open Science.

If the authors have any questions or comments regarding my peer review, please do not hesitate to contact me.

With kind regards,
Olmo van den Akker (ovdakker@gmail.com)

Reference

Bakker, M., van Dijk, A., & Wicherts, J. M. (2012). The rules of the game called psychological science. *Perspectives on Psychological Science*, 7, 543–554. doi:10.1177/1745691612459060

Appendix B

To the editor:

Thank you very much for your positive response to our manuscript, “Open science and modified funding lotteries can impede the natural selection of bad science.” We have revised the paper based on your feedback and that of the reviewers, and we hope the attached submission is worthy of publication in *Royal Society Open Science*. Below we document our responses to the reviewers’ comments. Original editor and reviewer comments are in **black**, our responses are in **blue**.

Associate Editor Comments to Author (Professor Zoltan Dienes):

I agree with the reviewers that your paper makes a clear and interesting contribution; I think addressing the impact of different funding schemes of scientific integrity is also timely. I think the manuscript stands clearly as it is; nonetheless, the reviewers provide thoughtful comments to take on board for a revision.

Thank you for this positive response. We received substantial peer feedback on our manuscript before submitting it, which contributed to its strength. We have endeavored to address the reviewer comments to the best of our ability.

Reviewer: 1

Comments to the Author(s)

This study builds from previous work to show that practices that reward quantity instead of quality of scientific publishing risk promoting bad science. The efficacy of science as a collective endeavor then suffers, both in the sense of making the published literature less reliable, and in the sense of rewarding scientists whose practices are more apt to generate erroneous results. These results help build the case for adopting open science practices and valuing negative results.

At an operational level, the manuscript does everything that a good modeling study should do. The model and its assumptions are clearly articulated, an interesting question is posed, and the question is explored thoroughly and honestly. Limitations of the model are discussed plainly, and interesting subsequent lines of investigation proposed. All of the ingredients that one expects are here, and the reader will have no trouble parsing the manuscript and understanding the essential results. The manuscript is also written with a satisfying stylistic flair that is rare among scientific articles.

Thank you very much for these positive comments. This last paragraph in particular was a delight to read about one’s work.

The relevance of the model hinges on whether the assumptions on which it rests faithfully reflect the actual state of affairs in science. To be sure, science is so vast that any attempt to capture some aspect of how the process works will invariably require strong assumptions. In these early days of science as a focus of inquiry itself, it is healthy to have a variety of models of science in the literature for readers and investigators to contemplate. Whether the assumptions on which this particular model is based are justified or accurate is fodder for debate, but such debate is healthy and drives the field forward. While I don't agree with every assumption here, a case could certainly be made in support of all the major assumptions on which this model rests, and that is all that one can reasonably ask.

For what it's worth, I would argue for a base rate higher than 10%, but I appreciate that the additional simulations in the Supplement demonstrate that the key results are robust to base rates across a large range. Also, though it is not central to the manuscript's essential findings, I wouldn't agree that the primary role of peer review is to weed out erroneous conclusions. One can hope that peer review will weed out the most egregiously flawed work, but most flawed papers are not so obviously wrong that a dedicated and conscientious reviewer could, by sheer strength of intellect, notice. Many of the practices discussed here under the banner of effective peer review (registered reports, double-blind review, improved statistical training) are entirely worthwhile, so perhaps my objection lies with the shorthand of labeling this collection of practices as "effective peer review". Nevertheless, I fear that the role of peer-review is often misunderstood, so would hesitate to further the notion that the primary role of peer review is to weed out erroneous results.

Again, thank you for your positive feedback. We believe that the base rate is often quite a good deal *lower* than 10%, particularly when one considers that every possible association or interaction tested is a distinct hypothesis. That said, we know that this point is contended, and which is why we provided the supplementary analysis.

We certainly agree that peer review can and should do much more than weed out erroneous results! We have added material in our Introductory section on peer review that acknowledges this, and notes that we are focused only on its corrective role in our analysis (end of section 2.2).

These are small quibbles, though. Overall, the manuscript clearly passes the standard for publication in this journal in its current form. I congratulate the authors on a nice contribution.

Thanks very much!

Reviewer: 2

In this paper, the authors use computational modeling to assess whether open science

improvements and funding lotteries are likely to improve the reproducibility of science. More specifically, they investigate the influence of three key factors: the publication of negative results, improved peer review, and the criteria for funding allocation.

I think that the authors make an important contribution by extending the 2016 model of the natural selection of bad science to include funding allocation and open science improvements. The inclusion of funding allocation is important because the 2016 model assumed that the rate at which research groups could produce results is limited only by the rigor of their methods. In my view, this assumption does not map well onto scientific reality and the availability of funding is a much more realistic way to model the potential output of research labs. The inclusion of open science improvements is important in the light of the increasing adoption of open science initiatives like preprints, open science badges, and registered reports. Although it makes intuitive sense that these improvements help us get to a literature with less false research findings, studies assessing the efficacy of these initiatives are currently lacking. While the current paper does not provide empirical evidence towards the efficacy of these initiatives, it does provide a model that makes valuable suggestions about which of the initiatives would be most worth pursuing. In all, extending the 2016 model this way has important benefits.

Even though I think the paper has enough value to be published as it is, I do have some comments and suggestions that may improve the quality of the paper. I discuss the points in order, but I must emphasize that I think the points related to the model itself are most important.

We thank the reviewer for this warm response. We worked hard to ensure our original submission was of high quality. The reviewer's additional suggestions are quite welcome, and we have done our best to address them.

Points related to the Introduction:

- 1) On page 5 the authors state that open science developments and changes to funding schemes can occur rapidly. Technically this is true as institutions can quickly change their policies. However, whether they will do so depends largely on cultural norms. Because cultural change can take a while, I would recommend the authors to be more nuanced in their discussion of the pace with which these developments can come about. Rephrasing the term 'rapid institutional changes' can go a long way here.

The reviewer is correct about the relationships between institutional change and cultural norms. We actually discuss this nuance at length in the paper's Discussion.

- 2) On page 6 the authors point out that the publication of negative results is getting more popular. The authors only mention registered reports as evidence for this development, but they could also mention the increased popularity of large-scale replication projects like Many Labs and RP:P. In addition, the authors mention some

journals that emphasize their willingness to publish null results on page 5, but I feel that that passage would be more suitable in this section.

We now also mention how journals are more willing to publish null results in this section as suggested. We are big fans of projects like Many Labs and RP:P, but these are highly specific large-scale replication efforts, and to our minds don't speak to a more general tendency to publish negative results.

- 3) On pages 6-9 the authors present several questions related to the extensions of the 2016 model. These questions appear to be the authors' main research questions. However, these questions are only implicitly answered in the results section. Explicitly restating and answering these questions would provide more structure to the paper.

We appreciate this suggestion. However, the questions posed in these pages are "big" questions, and providing definitive answers to them is beyond our ability with this paper. Their inclusion early in the paper was intended to steer the reader's framing of our model and interpretation of our results. Moreover, because our results are structured by topic subheadings that summarize the main results, we don't believe any additional structuring is necessary.

Points related to the model itself:

- 1) On page 10 the authors mention two hybrid strategies they included in the model: MS and ML, which are combinations of MI and RA, and RA and a threshold value of the false positive rate respectively. It makes sense to include such hybrids, but it is unclear why the authors only choose to implement these hybrids in the model. Why do the authors not incorporate a mix between PH and RA, and PH and MI, or even a mix of all three pure strategies? Some elaboration on the authors' choices would be welcome here.

We show in our simulations that there is negligible difference in the outcomes from PH and RA, which makes these suggested additional analyses minimally informative. More importantly, we focused on a mix between random allocation (RA) and methodological integrity (MI) because such a mix most closely approximates the recent calls for modified funding lotteries we document. We believe our motivation for the "hybrid strategies" we explore is well explained.

- 2) On page 12, Figure 1 indicates that the order of the stages in the model is 1) Science, 2) Evolution, 3) Grant-seeking. However, in my opinion it would make more sense if Grant-seeking is the first stage because labs can only do science if they have previously been awarded grant money.

As we explain, agents in our model are initially endowed with a "startup" fund, allowing them to conduct research without first securing a grant. This is to provide funders with some information about the activities of labs before making funding decisions. After this

initial stage, the phases of Science, Evolution, and Grant-seeking are cyclical, so this concern is a non-issue.

3) On page 12/13, the authors explain that a lab with enough funds will select a hypothesis to investigate in the Science phase. This hypothesis is true with a probability of b (the base rate of the field). In the model, b is an exogenous variable that does not change throughout the simulation. However, in reality, the types of hypotheses that labs choose may be highly dependent on the incentive structure in their field. For example, if novelty is the main funding criterion in the field, labs will look for hypotheses that are novel but unlikely to be true instead of hypotheses that are unimaginative but are likely to be true. As the model involves different incentive structures I think it would make sense to include b as an endogenous variable in the model.

The reviewer is correct that, in reality, the base rate reflects the ability of researchers to select true hypotheses, and thus varies between labs. However, the purpose of a model is not to accurately represent reality, but to make useful simplifying assumptions so that reality can be studied piecemeal. We have added a footnote to this section explaining this.

4) On page 13 the authors state that they assume a power of $W = 0.80$ in their model, but studies show that empirical power is much lower, at least in psychology (Bakker et al., 2012). It would be interesting to see the results of the simulation with a power of 0.30 or 0.40.

This is certainly true, which is why explicitly mentioned we were exploring an optimistic scenario with such a high value of power. More importantly, our analysis focuses on the qualitative changes to false discovery and false positive rates in response to selection pressures, which would be unaffected by differences in the assumed power. We did consider a more detailed discussion of power, but decided that it did not add to the paper and instead distracted from the model description.

5) On page 13 the authors note that in the 2016 model increased rigor not only yielded fewer false positives but also decreased the rate at which labs could produce new results and thereby submit new papers. In the current paper the authors disregard this assumption for reasons of tractability. Although I understand the need to avoid a model that is too complex, the authors do not provide a convincing argument why this assumption needed to be disregarded and not other parts of the model. An explanation and the presentation of results that do include this disregarded assumption would be welcome to assess the robustness of the findings.

We provide an explanation in the two sentences following our mention of this assumption: "Adding a reduction in productivity in response to rigor is likely to decrease the improvements from rapid institutional change. However, a theoretically motivated reason to ignore reduced productivity is an inherent difficulty in calibrating the extent to which such a reduction would manifest."

6) On page 14, the authors explain that in their model peer review works to keep false positives and false negatives out of the published literature. However, in reality, reviewers do not have accurate knowledge about the true state of the world and should not focus on the results of a paper but on its methodological rigor. Of course, reviewers currently do focus on the results of a paper, but they tend to focus on whether the results are positive and not on whether the results correspond to reality. In my opinion, it makes more sense for peer review to be operationalized similarly to the MI funding scheme. That is, peer reviewers would accept a paper for publication when it exceeds a threshold value of methodological integrity. This operationalization would also alleviate the need for the dubious assumption that peer review is equally capable of weeding out false positives and false negatives.

The reviewer appears to misunderstand the model. If peer reviewers worked as proposed, they would reject all papers by individual labs with a (deserved) reputation for decreased rigor. Moreover, their limited information is modeled by the probabilistic ability of peer reviewers to reject false discoveries – they are explicitly imperfect. Reviewer 1 expressed a related concern related to functions of peer review beyond reducing false discovery. As noted, we have now added a more nuanced discussion of the functions of peer review to the paper's introduction (end of section 2.2).

7) On page 14, the authors explain that in the Evolution stage each iteration a lab dies out and a new lab is created that adopts the methods of a progenitor lab. This means that the total number of labs in the simulation remains constant, which might not be in line with the fact that almost all fields of science have been increasing in the last few decades. Adding more labs instead of one may account for the constantly increasing size of scientific output.

It is true that many fields have grown (and a few have even shrunk). However, it is a common assumption in evolutionary modeling to use a fixed population size to focus on the relative frequencies of traits. Moreover, as we explicitly note in the paper's introduction, the number of job-seekers has grown faster than the number of job openings, which if anything shrinks the effective size of a field. In the absence of explicit hypotheses about the role of field size in the cultural evolution of scientific norms, it is a more prudent modeling assumption to keep the population size fixed.

8) On page 14, the authors explain that in the Evolution stage the false positive rate of a newly created lab is determined by the false positive rate of their parent lab and a random mutation. This mutation is truncated to only be positive, which might make sense given that new labs might not have as much methodological experience as older labs. However, the authors do not explain this choice, so the reader is left wondering.

"Methodological experience," is not actually a construct in our model. The false positive rate is inherited from a lab's "parent" lab, with unbiased error. This error, the mutation, is drawn from a normal distribution with a mean of zero (and so can be positive or negative). The resulting false positive rate (the inherited value plus error) is truncated to

be positive because the idea of a negative false positive rate is not mathematically coherent.

9) On page 16, the authors state that each simulation was run for 10 million iterations to ensure convergence to a stable mean false positive rate. This number of iterations is very large and one can doubt whether this large number is applicable to the actual evolution of science. If we suppose that every month somewhere in the world a research lab perishes and one is created, the number of iterations in the model would amount to more than 800.000 years of scientific evolution. If we suppose that it occurs every day, the number of years would still be more than 27.000. These are unrealistic timescales given that science as we know it is at most several hundred years old. Therefore, the large number of iterations in the model raises the question whether conclusions drawn from this simulation can be applied to the state of science in 10 or 20 years, which seems to be a main goal of the authors. I would be curious to see the results if the authors presented them on a more realistic timescale.

Figure 2 shows that most runs converged much more quickly, on the order of 10^5 iterations. We now note this explicitly in section 3.4. This estimate places the reviewers calculations on the order of something more like 270 years, which we view as not unreasonable. We now note more explicitly in this section that an iteration is not presumed to represent any specific length of time---our purpose is instead to illustrate more generally how selection works under our model's assumptions.

10) On page 15, the authors explain that it is randomly decided which lab applies for funding. However, applying for funding is not a random process as labs who focus on research / publishing papers have less time to apply for grants. One way this non-randomness could be incorporated in the model would be to only let labs apply for a grant if they have (nearly) run out of grant money. In my view, this would match reality better than the current model.

An earlier version of our model implemented funding in a manner very similar to that proposed by the reviewer. However, it was deemed overly complicated. We must remind the reviewer that the purpose of a model is not to accurately represent all aspects of reality, but to simplify reality so that factors unimportant to the dynamic under consideration can be safely ignored. We view this decision in that light.

Point related to the Results:

1) On pages 16-26, the authors explain in several sections what happens in the model when certain parameters are tweaked. However, the explanations for section 4.2 and 4.5 especially are not entirely clear to me. I think the Results section could benefit from a phase-by-phase description of the developments that are shown in the related figures.

We actually intended our presentation of the Results to be taken as a gradual increase in complexity, laid out in a clear, piecewise manner. We are sorry the reviewer did not

view it thusly, but without more specific examples, we are hard pressed to implement changes.

Points related to the Discussion:

1) On page 28, the authors state that “Such a model may indeed be a good representation of some modern academic disciplines”, but they do not state which disciplines this would be and why such a model would represent them well.

This is correct. We have chosen not to speculate on the extent to which this model represents particular disciplines, but instead simply pose it as food for thought.

2) Throughout the paper, the authors sometimes refer to replication and preregistration, but the authors do not specifically model replication, preregistration, or registered reports. Modeling these developments specifically would be welcome as they are very much at the forefront of current scientific infrastructure. Maybe the authors could indicate in the Discussion that a formal model including these developments could be a valuable research line.

We agree that some explicit modeling of these developments could be valuable. However, we don't feel the need to discuss this explicitly. We were very careful to use those examples only rhetorically, and to be clear about how their consequences were modeled.

Points related to language/clarity:

1) On page 4 the authors note that “the computational model presented by Smaldino & McElreath (2016) made several pessimistic—if realistic—assumptions about the ecosystem of academic science. We focus on two. First, it was assumed that publishing negative or null results is either difficult or, equivalently, confers little or no prestige. Second, it was assumed that publishing novel, positive results is always possible. In other words, the model ignores the corrective role of peer review or, equivalently, assumes it is ineffective.” To me, it is not directly clear how peer review is related to the prestige associated with publishing positive and negative results. One or two additional sentences explaining this link would be welcome.

From our perspective, it doesn't matter if negative results are published if they are not weighted similarly to positive results in decisions related to hiring and promotion. Thus, the important consideration is how null results “count” compared to positive results. We had made this clear in our Results section, but we agree with the reviewer that it could be made earlier. We have therefore added a sentence here to clarify this point.

2) On page 15 the authors use the term ‘false discovery rate’ to mean false positive and false negative findings in the literature. This term may be confusing because 1) its similarity to the term false positive rate, and 2) the fact that discovery implies a positive result (i.e., in common parlance, one cannot discover the absence of an effect). One

option to relabel these terms would be 'lab false positive rate' and 'false publication rate', but of course there are many options.

We are sympathetic to this concern, but choose not to change our wording, for several reasons. First, we are quite clear in how we operationalize the terms, and they are consistent both with our prior usage (Smaldino & McElreath 2016) and how the terms are generally used in other work. Second, we are not convinced that 'lab false positive rate' and 'false publication rate' are any more distinct. Third, we repeatedly explain why the quantities are different throughout the paper. And fourth, we disagree with the reviewer's assertion that one cannot discover the absence of an effect. To take famous examples from the history of physics and chemistry, demonstrations of the absence of phlogiston and the luminiferous aether were certainly discoveries.

3) On page 23 the authors mention: "Based on our finding that the effects of publishing negative results and improved peer review were cumulative, we use $p = r$ for simplicity". To me it is unclear what the authors mean by cumulative in this sentence and why it would lead to the assumption of $p = r$. This assumption would imply that the publication rate for negative results is equal to the efficacy of peer review, which does not seem to be well-aligned with reality.

By cumulative, we meant "additive" – i.e., they don't interact nonlinearly. We certainly don't assume p must equal r , but rather use their equivalence to reflect a more general extent of "open science improvements." We have edited this section to clarify this point.

4) On page 13 and 14 the authors mention that several additional robustness checks lead to "qualitatively similar" results. However, the authors do not specify what they mean by this phrase. Given that robustness checks are crucial in this kind of computational modeling, a specification of the qualitative nature of the results is desirable.

In all these instances we refer the reader to analyses done in the SI Appendix, in which the exact nature of these results is made quite explicit.

In all, I think the author's manuscript adds value to the current literature and provides us insight in the role of funding schemes and open science practices in attaining a more reproducible science. I do think that the model employed in the manuscript could use some additional parameter clarifications and robustness checks, but if those are implemented I would be happy to see the manuscript published in Royal Society Open Science.

We are glad that the reviewer finds value in our manuscript.